

# A 2000-year temperature reconstruction on the East Antarctic plateau, from argon-nitrogen and water stable isotopes in the Aurora Basin North ice core.

Aymeric P. M. Servettaz[1,2], Anaïs J. Orsi[1,3], Mark A. J. Curran[4,5], Andrew D. Moy[4,5], Amaelle Landais[1], Joseph R. McConnell[6], Trevor J. Popp[7], Emmanuel Le Meur[8], Xavier Faïn[8], and Jérôme Chappelaz[8]

[1]Laboratoire des Sciences du Climat et de l'Environnement, LSCE/IPSL, CEA-CNRS-UVSQ, Université Paris-Saclay, Gif-sur-Yvette, 91190, France
[2]Biogeochemistry Research Center, Japan Agency for Marine-Earth Science and Technology, Yokosuka, 237-0061, Japan
[3]Department of Earth, Ocean and Atmospheric Sciences, University of British Columbia, Vancouver, V6T 1Z4, British Columbia, Canada
[4]Australian Antarctic Division, Kingston, 7050, Tasmania, Australia
[5]Antarctic Climate and Ecosystems Cooperative Research Centre, University of Tasmania, Hobart, 7000, Tasmania, Australia
[6]Division of Hydrologic Sciences, Desert Research Institute, Reno, 89512, Nevada, USA
[7]Niels Bohr Institute, University of Copenhagen, Copenhagen, 2200, Denmark
[8]University Grenoble Alpes, CNRS, IRD, Grenoble INP, IGE, 38000 Grenoble, France

*Correspondence to*: Aymeric P. M. Servettaz (servettaza@jamstec.go.jp)

**Abstract.** The temperature of the earth is one of the most important climate parameters. Proxy records of past climate changes, in particular temperature, are a fundamental tool for exploring internal climate processes and natural climate forcings. Despite the excellent information provided by ice core records in Antarctica, the temperature variability of the past 2000 years is difficult to evaluate from the low accumulation sites in the Antarctic continent interior. Here we present the results from the Aurora Basin North (ABN) ice core (71° S, 111° E, 2690 m a.s.l.) in the lower part of the East Antarctic plateau where accumulation is substantially higher than other ice core drilling sites on the plateau, and provide unprecedented insight in East Antarctic past temperature variability. We reconstructed the temperature of the last 2000 years using two independent methods: the widely used water stable isotopes ($\delta^{18}O$), and by inverse modelling of borehole temperature and past temperature gradients estimated from the inert gas stable isotopes ($\delta^{40}Ar$ and $\delta^{15}N$). This second reconstruction is based on three independent measurement types: borehole temperature, firn thickness, and firn temperature gradient. The $\delta^{18}O$ temperature reconstruction supports stable temperature conditions within 1°C over the past 2000 years, in agreement with other ice core $\delta^{18}O$ records in the region. However, the gas and borehole temperature reconstruction suggest that surface conditions 2°C cooler than average prevailed in the 1000–1400 CE period, and support a 20th century warming of 1°C. These changes are remarkably consistent with reconstructed Southern Annular Mode (SAM) variability, as it shows colder temperatures during the positive phase of the SAM in the beginning of the last millennium, with rapidly increasing temperature as the SAM changes to the negative phase. The transition to a negative SAM phase after 1400 CE is however not accompanied by a warming in West Antarctica, which suggests an influence of Pacific South American modes, inducing a cooling in West Antarctica while ABN is warming



after this time. A precipitation hiatus during cold periods could explain why water isotope temperature reconstruction underestimates the temperature changes. Both reconstructions arguably record climate in their own way, with a focus on atmospheric and hydrologic cycles for water isotopes, as opposed to surface temperature for gases isotopes and borehole. This study demonstrates the importance of using a variety of sources for comprehensive paleoclimate reconstructions.

**Key Points:**

- Temperature reconstructions from water isotopes and borehole plus gas isotopes from the same ice core East Antarctica give substantially different climate histories for the past 2000 years.
- Water isotopes show low centennial variability, similarly to other East Antarctic Plateau ice cores; Borehole temperature and gas isotopes suggest a 2°C cooler 1000–1400 CE period, consistent with Southern Annular Mode
variability, as well as a 20[th] century warming of +1°C
- Differences emerge from the acquisition of temperature signal in the proxies, both spatially (atmosphere vs surface snow) and temporally (precipitation events vs diffusion over a few decades)

## 1 Introduction

The Antarctic continent is the only region where the recent warming trend cannot be distinguished from the high natural
climate variability (Abram et al., 2016). The sparse continuous measurements and the short timeframe covered by satellite era are insufficient to fully represent the climate in the vast continent that is Antarctica, and current climate models do not represent the full range of natural variability, highlighting that processes involved in the natural variability of temperature are still unclear (Jones et al., 2016). Therefore, climate archives such as ice cores can provide valuable information and help track the past evolution of temperature and climate (e.g., Jouzel et al., 2007; Stenni et al., 2017, among others). Understanding the
temperature variability in this ice-covered continent can improve the modelling of ice dynamics, melt events in coastal regions, predictions of ice sheet stability and the contribution to sea level rise, and is thus of critical importance when evaluating the risks associated with ongoing climate change (Meredith et al., 2022; Stokes et al., 2022).

The evolution of temperature during the last 2000 years is especially important as it provides a context of natural climate variability on timescales comparable to the recent warming, and temperature information is relatively well preserved in climate
proxies. Reviews of available temperature reconstructions were detailed for land (PAGES 2k Consortium, 2013) and oceans (Tierney et al., 2015). In Antarctica, the trends of different sub-regions for the past 2000 years were evaluated from temperature reconstructions based on water isotopes ($\delta^{18}O$ or $\delta D$) in ice cores, showing a general cooling (Stenni et al., 2017). The recent warming over the last 100 years is however not perceptible in all parts of Antarctica, resulting in an absence of significant warming for the continent as a whole. Ice cores included in the database used by Stenni et al., (2017) are unevenly spatially
distributed, especially the records covering the full 2000-year period, as ice cores are often drilled on domes or divides to avoid the glaciological advection of ice, leaving a vast gap of undocumented areas between coastal domes and the plateau summits.



In particular, the East Antarctic plateau temperature reconstructions are limited by low temporal resolution caused by the low accumulation in the dry high-elevation sites. In addition, most temperature reconstructions in Antarctica rely on water-stable isotopes, which could be seasonally biased (Werner et al., 2000; Persson et al., 2011), or altered by post-deposition effects (Landais et al., 2017; Casado et al., 2018). The regional climate would be better understood with temperature records from new locations, to increase the spatial coverage, and more diverse temperature proxies (Christiansen and Ljungqvist, 2017).

Studies of temperature variability in Antarctica have relied on temperature proxies from ice cores, especially water stable isotopes. The water stable isotopes measured in ice cores can be used to infer past temperatures, due to the relationship between cloud temperature and isotope composition of the precipitation: heavy isotopes in the atmospheric air mass are progressively flushed away as water condensates and precipitates (Dansgaard, 1964). Approaches to calibrate the isotope – temperature slope include: linear regression in the recent period when both temperature measurements and isotope records overlap (McMorrow et al., 2004; Steen-Larsen et al., 2014; Stenni et al., 2016; Casado et al., 2018), unidimensional isotope models (Ciais and Jouzel, 1994; Markle and Steig, 2022), or isotope-enabled general circulation model (Stenni et al., 2017; Goursaud et al., 2018). Even so, the slope calibrated over short periods may not transfer well to quantify the temperature variability at longer timescales (Jouzel et al., 2003; Casado et al., 2017), which is why the isotope – temperature slope should be carefully calibrated as close as possible as the expected variability (Jones et al., 2009).

Alternative solutions to reconstruct past temperatures have relied on the diffusion properties of temperature in the ice. Indeed, temperature of the ice is controlled by thermal diffusion and advection between the bedrock and the ice surface in contact with the atmosphere (Ritz, 1987). The bedrock-ice interface temperature, which depends on geothermal heat flux, is relatively stable at the timescales of a few thousands of years. Therefore, the atmosphere-surface snow temperature is the main source of variation in the ice temperature profile. This profile of temperature can be directly measured in the borehole after an ice core has been drilled, and can then be used to infer past surface temperature changes, using inverse methods and diffusion models with known heat diffusion properties in the ice (e.g., Johnsen et al., 1995; Dahl-Jensen et al., 1998; Orsi et al., 2012).

Finally, it is possible to assess the past temperature gradients from the diffusion of inert gases in the firn. The firn is the porous layer of compacted snow at the top of the ice sheet, which allows for movement of gases, mainly by diffusion. The diffusion of the isotopes of inert gases differs because of their physical properties, with primary control by gravitational settling of heavy gases at the bottom of the diffusive column (Craig et al., 1988; Sowers et al., 1992). In addition, a temperature difference between the two ends of the diffusive column can create an isotopic signal that can then be captured in the air bubbles trapped in the ice matrix (Severinghaus et al., 2001). The diffusive zone of the firn gases lies between the convective zone above, where gases are actively mixed by surface winds (Kawamura et al., 2013) or pressure changes (Buizert and Severinghaus, 2016), and a lock-in depth below, where the ice layers that merged under increased pressure block the vertical movement of gases (Buizert et al., 2012; Fourteau et al., 2019). Further below is the close-off depth where porosity is fully sealed, and gases are trapped in bubbles enclosed within an ice matrix (Severinghaus and Battle, 2006). The analysis of inert gas isotopes can thus be used to infer the past temperature gradients in the diffusive column of the firn (Kobashi et al., 2008). Reconstruction of temperature changes at the surface requires a thorough understanding of the vertical profile of temperature in the ice and





can be achieved by inverse modelling of a vertical temperature diffusion model in the firn (Orsi et al., 2014). Inversion of past temperature gradients estimated from gases isotopes has been applied in Greenland (Kobashi et al., 2008; Orsi et al., 2014), and some places in Antarctica (Orsi, 2013; Morgan et al., 2022), but remains rare compared to temperature reconstructions based on water isotopes, given the longer processing time, the volume of ice needed, and the analytical precision required in

inert gas analyses. Moreover, the accumulation rate controls the gases bubble enclosure speed, and restricts the locations where this method can be used to infer temperature changes: a too slow accumulation rate lets the time to the gases to diffuse out the temperature signals, while a too fast accumulation rate does not let the time for the gases to equilibrate with the firn, which is why this method is best used at sites within a 60–300 kg m$^{-2}$ yr$^{-1}$ accumulation range.

In short, the current temperature estimations for the East Antarctic Plateau are hindered by the low temporal resolution of ice

cores drilled in this region, which can be limiting to clearly assess the trends and variability at a sub-millennial timescale because of deposition dynamics and post-deposition processes masking the climate signal (Münch and Laepple, 2018; Casado et al., 2020). Moreover, the large majority of temperature reconstructions rely on water stable isotopes, in a region where ice accumulation is uneven through time (Turner et al., 2019) and may induce biases in water isotopes based reconstructions. In this study, we try to address both issues by reconstructing the temperature from the Aurora Basin North ice core with two

independent methods. We first use the water stable isotopes and expect that the relatively high ice accumulation for a Plateau site, of about 100 kg m$^{-2}$ yr$^{-1}$ (Akers et al., 2022), will result in a more detailed temporal resolution and may better record the centennial variability of East Antarctic Plateau temperatures. Then, we use gases isotopes and borehole thermometry in an inverse model to retrieve surface temperature changes. We finally compare the two reconstructions and discuss the climate implications along with other Antarctic records.

## 2 Material and methods

### 2.1 Aurora Basin North site description

Aurora Basin North (ABN) is located inland East Antarctica, at 71.17° S, 111.37° E at 2,690 m elevation. The ABN site is approximately mid-distance between the coast and Dome C, on the Indian Ocean sector (Fig. 1). This site is located on the East Antarctic plateau, but much closer to the ocean and thus receives significantly more snowfall than EPICA Dome C or

Vostok, making it more suited for studies of the late Holocene climate, with a water isotope record resolved at a near annual scale (Moy et al., 2017). The 20[th] century snow accumulation at ABN is estimated at $126 \pm 26$ kg m$^2$ yr$^{-1}$, up from the $94 \pm 18$ kg m$^2$ yr$^{-1}$ before 1900 CE (Akers et al., 2022). The temperature at ABN shows a high positive correlation with a large part of the Indian sector of continental East Antarctica at interannual scale, as estimated from a regional atmospheric model (Servettaz et al., 2020). A 303 m core was drilled at ABN in the summer of 2013–2014, named ABN1314, which is used in

this study. Additionally, we use data from a 12 m shallow firn core to cover the most recent snow (described in Servettaz et al., 2020).



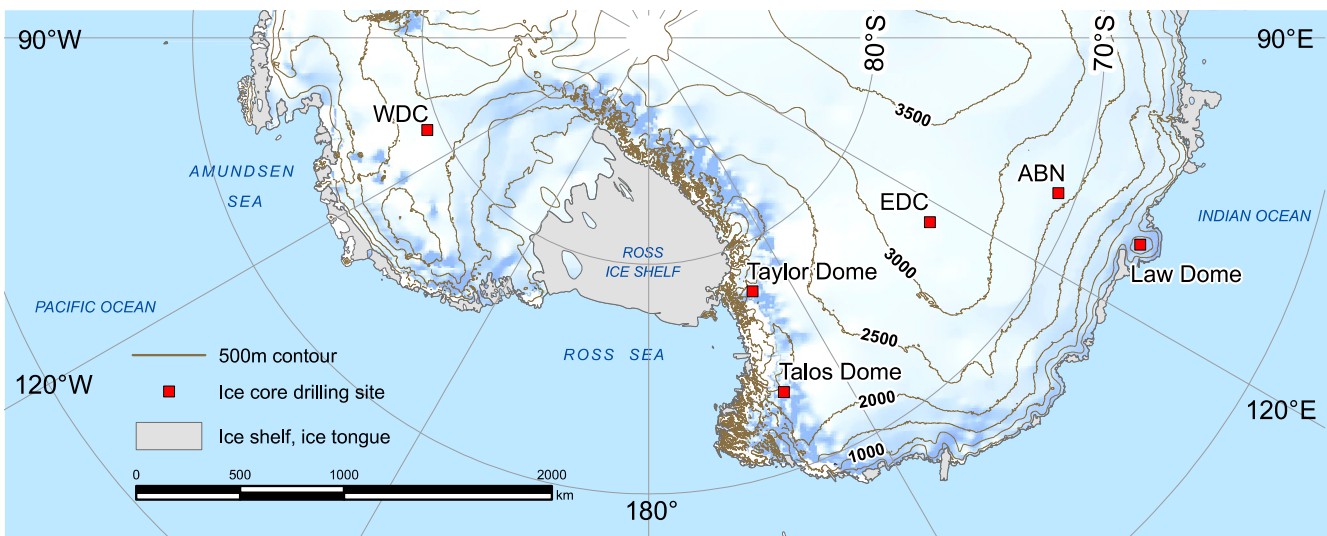

**Figure 1.** Map of the Indian and Pacific sectors of Antarctica. A selection of ice core drilling sites is shown: ABN – Aurora Basin North (this study), Law Dome, EDC – EPICA Dome C, WDC – West Antarctic Ice Sheet Divide Core, Talos Dome, Taylor Dome. True scale at 71° S. Modified from a production of the Australian Antarctic Data Centre, June 2009 (map catalogue no. 13641 – data.aad.gov.au © Commonwealth of Australia 2009). Creative Commons Attribution 4.0 Unported License.

## 2.2 Measurements

### 2.2.1 Water stable isotopes

The ABN1314 core was subsampled at 20 cm resolution for water stable isotope analysis at the Australian Antarctic Division. Water stable isotopes ($\delta^{18}O$ and $\delta D$) analysis were performed on a Picarro L2130-i isotopic water analyser. Aliquots of water were sampled by a Picarro liquid auto-sampler and injected into a Picarro high precision vaporization module (A0211) and held at temperature of 110°C with the vapour sent to the Picarro L2130-i isotopic water analyser. Isotopic values are expressed as per mil (‰) and relative to the Vienna Standard Mean Oceanic Water (V-SMOW) standard. The standard deviation of the $\delta^{18}O$ values for repeated measurements of laboratory reference water samples was less than 0.05 ‰ and less than 0.5 ‰ for $\delta D$. A total of 1522 measurements were performed on the 303 m core. Although $\delta^{18}O$ and $\delta D$ are roughly proportional, deviation from the standard meteoric water line likely results from changes in the evaporation conditions at the moisture source (Dansgaard, 1964; Uemura et al., 2008). The deviation from the meteoric water line can be defined with a logarithmic notation (Uemura et al., 2012):

$$d_{\ln} = 1,000 \times \ln(1 + \delta D) + 2.85.10^{-2} \times \left(1,000 \times \ln(1 + \delta^{18}O)\right)^2 - 8.47 \times 1,000 \times \ln(1 + \delta^{18}O) \qquad (1)$$

The propagated uncertainty gives an analytical precision for $d_{\ln}$ of 0.52 ‰.

### 2.2.2 Gases stable isotopes

Gas-dedicated samples of about 20 cm in length were subsampled approximately every 2 m in the ABN1314 core. Due to the porosity of the ice sheet, gas samples were taken below the lock-in depth, starting at 104 m depth. The samples' outer ice was



shaved off to prevent contamination by exchange with air during transport and storage, and each sample was split into two

duplicates of ~70 g each.

In order to measure precise argon isotopes, $O_2$ was removed due to isobaric interference between $^{36}$Ar and $^{18}$O$^{18}$O. Argon and nitrogen gases were extracted from the ice following the method of (Kobashi et al., 2008): the ice was melted in a pre-emptively evacuated bottle, and the gases were released in a processing line with cold traps to remove water vapour and carbon dioxide, and then uses a heated copper mesh (500°C) to remove molecular oxygen. The remaining gases are trapped in a collection tube

that is cooled with liquid helium. To maximize the efficiency of the traps, the gases were released slowly in the processing line, with the pressure in the collection tube maintained under 50 Pa at all times. Once the entirety of the gases was released into the processing line, the collection continued for about 15 minutes, until the pressure drops under $5 \cdot 10^{-2}$ Pa, to ensure complete trapping of the gases. The collected gases are left to heat up to room temperature and homogenise in the collection tube overnight, and then measured on a dual inlet mass spectrometer (MAT 253+) against a laboratory standard of $N_2$, Ar, Kr.

The MAT 253+ spectrometer was setup so that different isotopes of a same element are measured simultaneously in an arrangement of collection cups dedicated to $N_2$, Ar, or Kr. The standard is calibrated weekly against modern air following the same protocol as the ice sample, from release in the processing line to mass spectrometry.

We use the $\delta^{40}$Ar and $\delta^{15}$N notations for isotopic ratios $^{40}$Ar/$^{36}$Ar and $^{15}$N/$^{14}$N in the sample relative to the isotopic ratios in the free atmosphere (IAEA, 1995). The dual inlet system that sends gas to the spectrometer switches between the sample and

laboratory standard in cycles for robust measurement of the isotopic ratio. Cycles of 11 standard and 10 sample injections were repeated 5 times with the spectrometer in argon configuration and 3 times in nitrogen configuration. Additionally, elemental ratios were measured following the peak-jumping method (Bereiter et al., 2018).

Pressure imbalance and chemical slope corrections were applied to account for the imbalance in the dual inlet system and the elemental ratios in the sample, first described by Severinghaus et al. (2003; details of the corrections used in this study are

given in Appendix A3). In addition to the previously described corrections, it was identified that the spectrometer focus drifted over time, that resulted in a strong variability of both $\delta^{40}$Ar and the value used pressure imbalance correction. While the measurements took about nine months to complete, the pressure imbalance correction value was verified daily, and estimates of the intensity of $\delta^{40}$Ar error resulting from the spectrometer focus were made using the daily value of pressure imbalance correction, which is itself dependant on spectrometer parameters. We thus corrected for the spectrometer drift (details of this

correction are given in Appendix A3). This drift correction halved the pooled standard deviation of $\delta^{40}$Ar in the ice duplicates. While Kobashi et al. (2008) have reported argon loss during storage of ice for extended amount of time at a temperature of −20°C, the excellent quality of ice from a recently drilled ice core, and the precautions taken during the preparation prevented any notable effect of argon loss during storage on the $\delta^{40}$Ar measured in our samples.

A total of 102 pairs of duplicates were analysed, and this was complemented by three single samples that could not be

duplicated because they were taken from ice samples that were too small to be split in two, or for which one sample of the duplicate pair was lost due to a leak during processing. Given ice samples were processed in duplicates and have undergone the same procedure, we evaluate the reproducibility of the method and analysis by comparing duplicate results. The pooled





standard deviation of the 102 duplicates is 0.0159 ‰ for $\delta^{40}$Ar and 0.0045 ‰ for $\delta^{15}$N. Three samples had exceptionally high duplicate differences, with higher than three times the pooled standard deviation for either $\delta^{40}$Ar or $\delta^{15}$N. These samples had

been flagged as potentially contaminated due to a change of operator and an unplanned power outage during processing. Another four shallow duplicates, taken just below the close-off depth, had $\delta^{40}$Ar and $\delta^{15}$N values significantly lower than the gases in the open porosity of the lock-in depth above. At ABN, the close-off depth of ~104 m below surface is where the pores are completely closed, and the air bubbles are trapped in an ice matrix. Recent bubbles in the ice around this depth were probably thinly closed at the time of sampling and may have been contaminated with more recent air before the pores were

fully sealed. Another possibility is that vacuum pumping on thinly closed porosity may have altered the isotopic composition during gas sample preparation. These samples were considered as outliers and will not be included in the final dataset, which consists of 95 duplicates and 3 singletons. After removal of outliers, the pooled standard deviation of remaining samples is 0.0137 ‰ for $\delta^{40}$Ar and 0.0036 ‰ for $\delta^{15}$N.

The bubble trapping was shown to be heterogeneous at the 10 cm scale, causing variability in the isotopic composition of the

gases (Orsi, 2013), probably because of the differential closure rate of bubbles in summer versus winter layers of ice (Severinghaus and Battle, 2006). Due to the high frequency variability of gases, the isotopic composition cannot be related to climate information. The data is thus smoothed by resampling using a 5 m window to average both $\delta^{15}$N and $\delta^{40}$Ar (Fig. 2). The uncertainty on the resampled points was inferred as the standard deviation of all points in the 5 m window. Hereafter in this study, the smoothed data with a 5 m resolution is used.





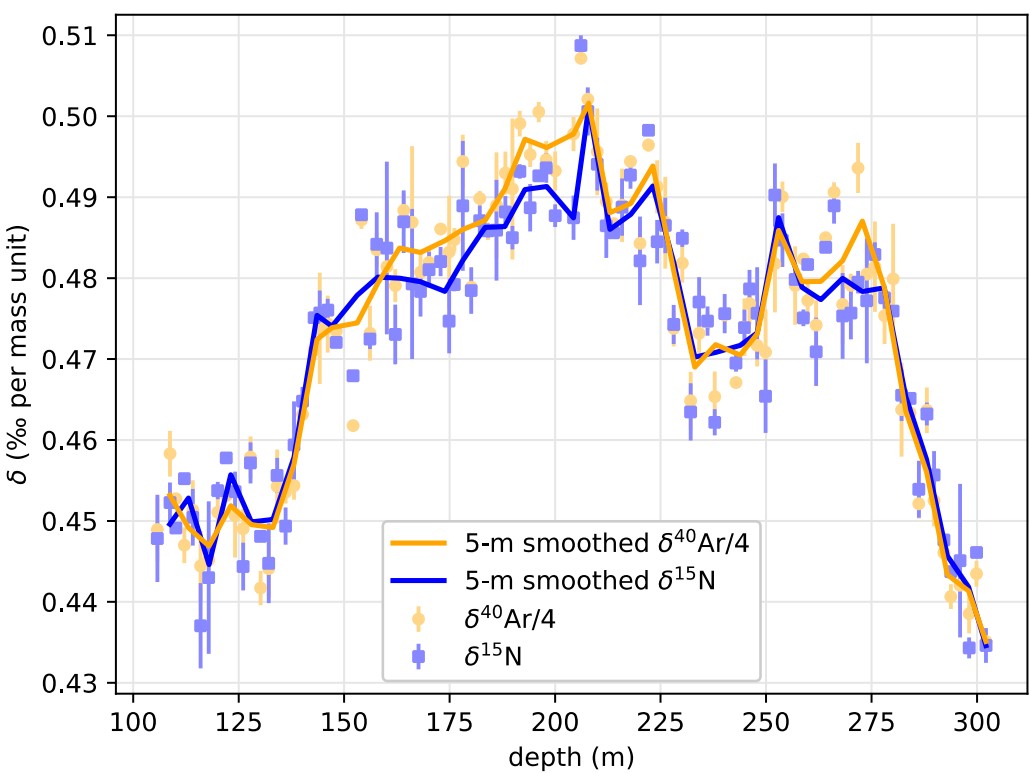

**Figure 2. Series of δ⁴⁰Ar (orange) and δ¹⁵N (blue) in the ABN1314 core. Large dots and squares represent the average value for a given depth, error bars illustrate the difference between duplicates. The 5 m smoothing is shown with solid lines for both series.**

### 2.2.3 Borehole temperature

The Borehole temperature measurements were made by lowering a measuring resistor with a three-bladed probe down the borehole. The resistance was measured using a Fluke multimeter, and converted to temperature using temperature dependence of resistivity. The probe was stopped at target depths and let equilibrate with surrounding ice until a stable measurement could be achieved. A second series of measurements was made at the same depths when the probe was pulled back up, providing two temperature values for each depth (downward and upward, Fig. 3). Wet drilling (Estisol) commenced from 132 m, and it is very likely that the open markers in Fig. 3 are outliers due to disturbance of air in the drill hole with warm fluid stored at the surface. Below 132 m, the small difference between upward and downward measurements is likely due to improved equilibrium in the drilling fluid. Relatively short equilibrium time before each measurement may have caused the downward temperature to be overestimated whilst the upward temperature series is underestimated because of the memory effect of the temperature probe, particularly above the drilling fluid. Even though the temperature was measured only few weeks after the drilling had completed, the heating effect of the drilling is relatively small, estimated at ~0.1°C (Orsi et al., 2017). The



temperature measured ranged from –45.5°C at the bottom of the drill hole to –43.1°C at 17 m below the surface, deep enough

for the temperature not to be affected by seasonal variations, revealing a gradient larger than 2°C over the 300 m depth.

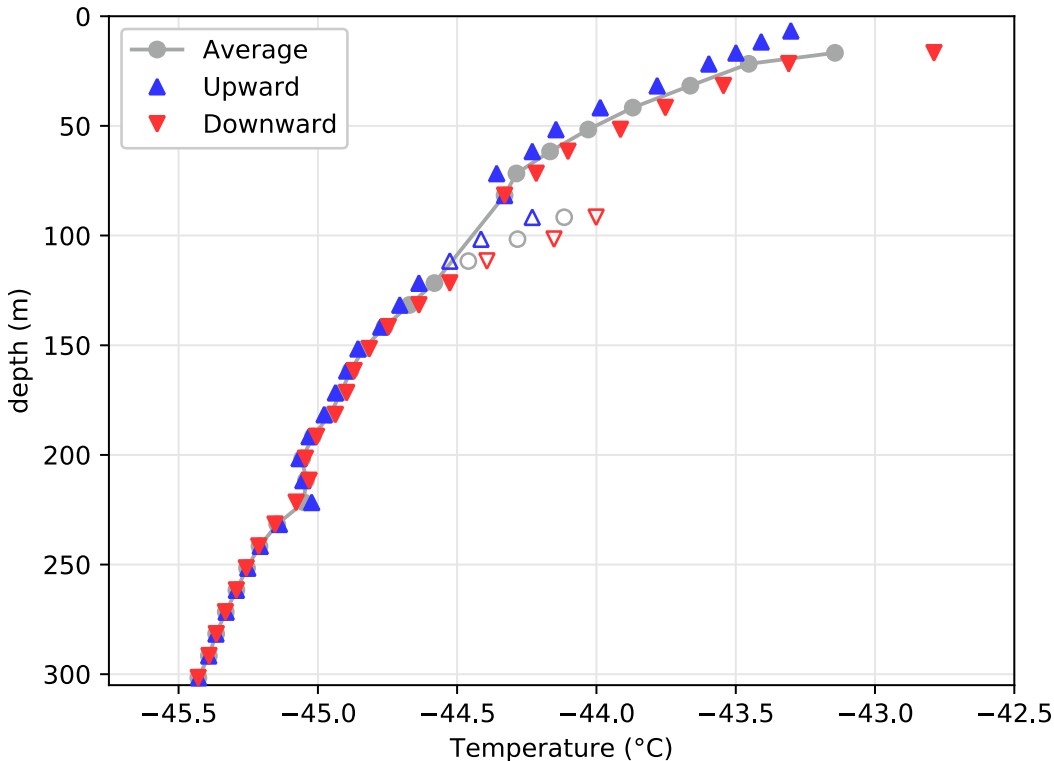

**Figure 3. Temperature profile in the Aurora Basin North main core borehole. The drilling fluid may have induced a disturbance of air temperature in the drill hole because the drilling fluid was stored at the surface at a warmer temperature of about –20°C. Open**
**markers indicate points that will be ignored for this reason. Below 132 m, the small difference between upward and downward measurements is likely due to improved equilibrium in the drilling fluid.**

### 2.3 Ice core age model

### 2.3.1 Ice age model

Producing a well-constrained age model is a necessary step in the analysis of a paleoclimate record. Chemical species, electro-
conductivity, and water stable isotopes were measured at high resolution with Continuous Flow Analysis (CFA) at the Desert

Research Institute (Maselli et al., 2013; McConnell et al., 2002), allowed identification of annual cycles in most of the

ABN1314 ice core. The ABN1314 core was dated by Annual Layer Counting (ALC) of seasonally varying aerosols ($Na^+$, Cl),

electro-conductivity and water isotope measurements. The ALC was performed manually and subsequently tied to volcanic

events using sulphate aerosols from other well dated ice core records. The ABN1314 total sulphur record was compared to the
Plateau Remote (Cole-Dai et al., 2000) and West Antarctic Ice Sheet Divide (Sigl et al., 2013) ice cores, as all sites are inland



Antarctica and have similar backgrounds for sulphur. Even though Law Dome is the nearest site with high resolution sulphur record, the background is very different, likely due to Law Dome being a coastal site. ALC was adjusted using 29 volcanic horizons that were matched to the WD2014 chronology (Sigl et al., 2016). The surface snow mixing by wind at ABN is of comparable scale with the yearly accumulation (up to 40 cm of snow, Servettaz et al., 2020), which may have hindered the

identification of annual signals used for year identification in ALC especially during periods of lower accumulation. Therefore, the ABN1314 ALC was anchored to the WD2014 chronology with volcanic ties, and then ALC was performed a second time in between ties to find the expected number of years. The uncertainty on the age of volcanic horizons in the WD2014 chronology is lower than 5 years, and hence we report all the volcanic ties with a ± 5-year margin to account for this uncertainty. The ABN1314 core covers the last 2700 years (Fig. 4). Dating uncertainties resulting from the ALC were considered as follows:

if an annual layer was counted but is not on a clear seasonal extremum, it is flagged as uncertain, and a half-year uncertainty is added (details on the uncertainty is shown in Fig. A1). The age model for the ABN1314 core is referred to as ALC-01-11-2018. Through the volcanic age tying, the age model for ABN1314 has consistently low uncertainties of less than 20 years.

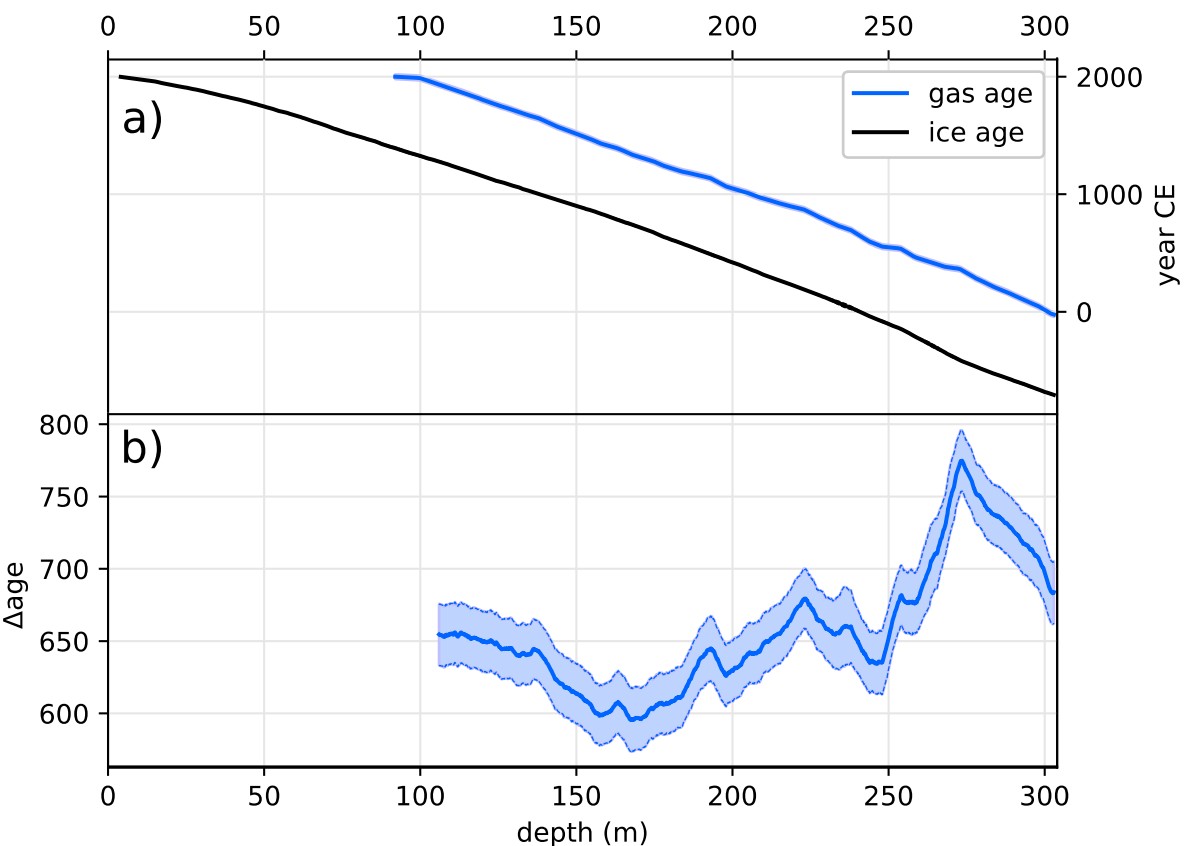

**Figure 4. (a) Ice and gas age models for the ABN1314 ice core. The ice age model is based on Annual Layer Counting of seasonally**
**varying chemical species, while gas age model is based on methane ties. Both age models are tied to WD2014 chronology for ice and**



**gases respectively (Sigl et al., 2015a, 2016b). (b) Δage defined as the difference between ice and gas ages at each corresponding depth. Shading indicates the estimated uncertainty.**

### 2.3.2 Gas age model

The gases are trapped by closure of the porosity at the bottom of the firn, and hence are enclosed in bubbles surrounded by
substantially older ice. Preliminary dating of the gases is completed by subtracting a constant age difference to the surrounding ice, corresponding to the modern-day difference between age of gases and ice at the lock-in depth. Firn characteristics may vary through time, affecting the height of the diffusive zone and thus the lock-in depth, and hence the gas age model is further refined with the methane record measured in the ABN1314 core.

The methane content of gases trapped in ABN1314 was measured with the Desert Research Institute Continuous Flow Analysis
system that was modified for gas measurement (Rhodes et al., 2013). We used methane records to tie gas ages scale of the ABN1314 core with the established chronology of the West Antarctic Ice Sheet (WAIS) Divide ice core, named WD2014 (Sigl et al., 2016). Specifically,  we defined tied points so that the methane content in bubbles at ABN fits the published methane dataset from the West Antarctic Ice Sheet (WAIS) Divide ice core, named WD2014 (Sigl et al., 2016). Tie points were identified where there is clear, quick transitions or extrema on methane records (Fig. A2).

For the most recent part (1800 to 2000 CE, Fig. 4b), the methane data from WAIS Divide was not available, and hence the ABN methane record was tied to the revised Law Dome record (Rubino et al., 2019). Again, samples taken between above the close-off depth of ~104 m below surface, which corresponds to 1925 CE, are likely to be contaminated with recent air before the pores were sealed, resulting in higher methane concentrations.

The gases are younger than the ice at the same depth, so the gas age model of ABN1314 core only covers the last 2050 years.
The most recent gases are technically still in the diffusive column of the firn, or partly in the remaining open porosity between lock-in and close of depth, but not fully trapped in bubbles within an ice matrix. The uppermost ice core samples that contain gases suited for analyses are found below the close-off depth, dated to 1925 CE. Because the ages were tied manually, it is difficult to estimate the uncertainty, but the tie points used correspond to events with an age span shorter than 20 years. While the lock-in depth of gases may vary depending on the species due to diffusivity speed in the firn (Witrant et al., 2012), the
resulting difference is expected to be much lower than 20 years at ABN. Therefore, we roughly estimate the uncertainty on the gas age model to be 20 years.

### 2.3.3 Gas-ice age difference

The difference between gas age and ice age provides information on the dynamics of the firn and its evolution through time. A larger age difference may result from a decrease in accumulation rate or an increase of lock-in depth where gases are trapped.
Fig. 4 shows the gas and ice age models and the difference of age between gases and ice at a given depth. The gas-ice age difference is comprised between 600 and 700 years, except for one excursion at around 270 m depth which is likely related to



the sudden change of accumulation caused by a dune-like feature upstream from the ABN site, which was advected under the current ABN location with ice flow.

## 2.4 Ice flow correction

In opposition to many ice core drill sites that target a dome or a divide, ABN was drilled in a basin setting, where ice slowly flows from the continent to the coast. The estimated ice flow is determined by comparing the accumulation record from the ice age model, and the first isochron reflector depth from the ground penetrating radar survey upstream of ABN: the first order of accumulation changes is driven by local accumulation features caused by the topographic slope (Van Liefferinge et al., 2021; Akers et al., 2022). Indeed, the speed of the katabatic wind scales with terrain slope (Parish and Bromwich, 1991; Vihma

et al., 2011), and thus acceleration of wind scales with terrain curvature. Accelerating winds can charge up in snow particles and locally reduce the accumulation. Conversely, decelerating winds deposit the drifting snow, increasing the accumulation. By matching the time-series of accumulation from the ABN1314 core to the upstream accumulation patterns (Fig. A3), we estimate that the bottom part of the ABN ice core corresponds to ice originating 41.5 km upstream from the ABN drill site, which results in an average ice flow of 15.4 m yr$^{-1}$. This value is in relatively good agreement with satellite-based estimation

of 16.2 m yr$^{-1}$ at ABN for the 1996–2018 period (Mouginot et al., 2019).

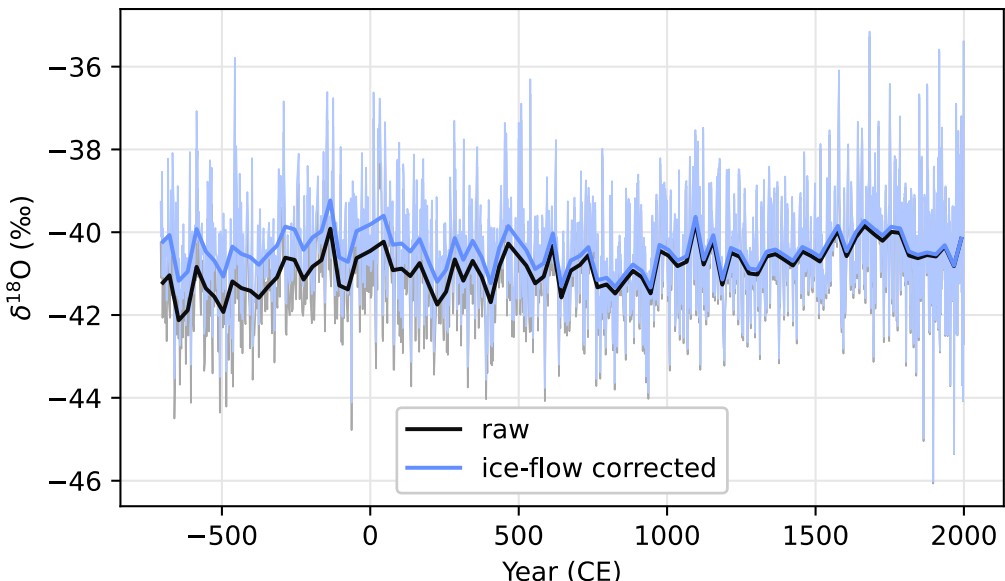

**Figure 5. δ$^{18}$O in the ABN1314 ice core, with a sampling resolution of 20 cm (thin lines) and on 30-year average (Thick lines). The analytical uncertainties on δ$^{18}$O of 0.5 ‰ are not shown on the figure. Black lines show the raw measurement and blue lines show the glaciology-corrected values.**

Using this ice flow velocity, the estimated origin altitude of the ice in the ABN1314 core, or the paleo-elevation of the ABN drilling site is made. The ice dating from 700 BCE was formed at an elevation of 2745 m, and subsequent ice forming the



ABN1314 ice core originated from lower elevation, down to the drill site at 2690 m for the most recent snow (Fig. A3). We estimated that a decrease of 55 m elevation would cause a temperature increase of 1°C, using the temperatures interpolated with kriging from Automatic Weather Stations and borehole temperature measurements for two traverses in nearby Princess

Elizabeth Land, East Antarctica (Xiao et al., 2013; Pang et al., 2015; Fig. A4). The regression was performed on a subset of the traverse between 2000 and 3250 m above sea level, corresponding to the lower plateau around the elevation where ABN is located. Similarly for water stable isotopes, the estimated decrease of 55 m elevation would result in a $\delta^{18}$O increase of about 1 ‰, by calibrating a $\delta^{18}$O–elevation slope in surface snow studies at elevations comprised between 2000 and 3250 m above sea level, and longitudes from 80° E to 160° E (Xiao et al., 2013; Pang et al., 2015; Goursaud et al., 2018; Fig. A5). The

measured $\delta^{18}$O and ice-flow corrected $\delta^{18}$O are shown on Fig. 5. We did not find any significant relationship between elevation and $d_{ln}$ at this elevation range (Fig. A6), and therefore did not correct it for glacial flow.

## 3 Temperature reconstructions

### 3.1 Water stable isotopes

Water isotopes in ice cores have consistently been used to estimate past temperatures (e.g., Cuffey et al., 1995; Jouzel et al.,

1997, 2003; Stenni et al., 2017). The ABN isotope record consists of 1522 data points for the 2707 years covered by the ice core, resulting in a temporal resolution of $\delta^{18}$O slightly above one point for two years. However, the water isotopes are strongly affected by deposition variability and non-climatic factors that may mask the climate signal at high resolution, especially for regions such as the East Antarctic Plateau where the accumulation is low (Münch and Laepple, 2018; Casado et al., 2020). Moreover, a few extreme precipitation events can be responsible for a large part of the total yearly accumulation (Turner et

al., 2019), so it is more reliable to average over several years to retrieve the climate information from the water isotope record. Therefore, a 30-year average is used to better determine the climate signal. The 30-year averaged $\delta^{18}$O series has a mean value of −40.8 ± 0.5 ‰ (1σ) and shows minimal variation over time.

The ABN $\delta^{18}$O – temperature slope is determined using the $\delta^{18}$O record from the 12 m shallow core that is dated back to 1968. The shallow core overlaps the time period with the Modèle Atmosphérique Régional (MAR) modelled temperature on the

1979–2013 period (Agosta et al., 2019), and MAR was shown to model the surface temperature more accurately than any other available dataset when compared with automatic weather station observations near ABN (Servettaz et al., 2020). MAR is thus used as the temperature reference for ABN, because the nearest automatic weather station temperature data record (1985–1993) is too short to capture the interannual variability of temperature. We calibrate a $\delta^{18}$O – temperature slope for ABN using linear regression on the 1991 to 2013 period, where we are confident on the dating and have a decent number of

years. Both temperature and $\delta^{18}$O are averaged annually to minimise the influence of seasonal variability.



**Figure 6.** Calibration of δ¹⁸O – temperature slope with the Desert Research Institute (DRI) Short Core δ¹⁸O and the 2 m temperature in the Modèle Atmosphérique Régional (MAR) on the 1991–2013 CE period. **(a)** Annually averaged series of MAR 2 m temperature (orange) and DRI Short Core δ¹⁸O, on their overlapping period of 23 full years from 1991 to 2013. **(b)** Scatterplot of the two datasets (black crosses), with linear regression in blue solid line. Dashed lines represent the slope upper and lower uncertainties at a 95% confidence level. **(c)** Temperature reconstruction using the δ¹⁸O series from the ABN1314 core and the slope obtained with linear regression, with temperature reconstructed with upper and lower uncertainty slope values in dashed lines.

We determine a $\delta^{18}O$ – temperature slope $\alpha = 2.01$ ‰ °C$^{-1}$, with a 95 % confidence interval of $1.16 < \alpha < 2.87$ ‰ °C$^{-1}$. According to this slope, a change of 1°C in the mean annual temperature would be recorded in the snow by a ~2 ‰ change in $\delta^{18}O$. This 2.01 ‰ °C$^{-1}$ slope is relatively high compared with other slope values estimated with ECHAM5-wiso, which average 1.00 ‰ °C$^{-1}$ in East Antarctica (Stenni et al., 2017), or 0.85 ‰ °C$^{-1}$ at ABN (Servettaz et al., 2020). This difference



could be at least partly attributed to the ECHAM5-wiso model, known to underestimate the inter-annual variability of isotopes in Antarctica (Goursaud et al., 2018).

Here, we use the 2.01 ‰ °C$^{-1}$ slope to convert the $\delta^{18}$O corrected for ice flow from ABN1314 to a reconstructed temperature record (Fig. 6c). The ice flow correction described in Sect. 2.4 is applied to the $\delta^{18}$O data before converting it to a temperature record, to emulate a $\delta^{18}$O record at the current ABN location. The $\delta^{18}$O–temperature calibration slope was defined on a shorter period of 23 years that should not be affected by ice flow. The temperature obtained from this reconstruction averages −42.0°C and remained within a 1°C range (from −42.6 to −41.6°C) over the past 2700 years (Fig. 6). The use of a different $\delta^{18}$O –

temperature slope slightly modifies the range of temperature values (Fig. 6c) but does not modify our interpretation. We acknowledge that using a $\delta^{18}$O – temperature slope calibrated with yearly averages is not optimal given the limitations caused by deposition dynamics and post-deposition processes, but the $\delta^{18}$O – temperature calibration is limited by the length of overlapping $\delta^{18}$O and available temperature records. The temperature reconstruction from $\delta^{18}$O at ABN might also be biased, as precipitations occur consistently during warm events (Servettaz et al., 2020).

**3.2 Gases and borehole temperature inversion**

While the water isotopes have been the proxy of choice for many paleoclimate studies in Antarctica, past temperatures are also imprinted in the ice and can be measured in the borehole after the ice core has been drilled. However, the temperature slowly diffuses in the ice, smoothing out the signal over time. Diverse inversion methods have been applied to estimate temperature changes since the last deglaciation (e.g., Dahl-Jensen et al., 1998) or over the last millennium (Orsi et al., 2012), but the

temperature reconstruction from borehole quickly loses temporal resolution over time. Another method to estimate past temperature changes relies on the thermal fractionation of gases in the firn diffusive column, where a gradient of temperature can be captured in the $\delta^{15}$N and $\delta^{40}$Ar composition of the air trapped in bubbles of the ice core (Kobashi et al., 2008; Orsi et al., 2014). Here, we also reconstruct past temperatures at ABN using a combination of the borehole temperature and the past temperature gradients estimated from gases isotopes.

**3.2.1 Determination of temperature profile and past temperature gradients**

The temperature profile was measured directly in the borehole after ice core was drilled. The temperature profile of ABN borehole (Fig. 3) shows a strong decrease of temperature at depth, suggesting that the surface warmed while old ice buried under remained colder. We observe a near-surface temperature gradient of ~1°C 100 m$^{-1}$, for which a 2.7°C warming over the last 30 years was inferred at a Greenland site (Orsi et al., 2017). On the East Antarctic Plateau however, no such warming

trend was previously detected (Nicolas and Bromwich, 2014). We expect that the ABN temperature gradient partly results from ice rheology, displacing the ice from a colder location upstream to what is now buried under ABN site.
Gravitational and thermal fractionation affect the gas contained in the firn porosity. The temperature gradient in the diffusive column controls the thermal fractionation, while the height of the diffusive column determines the gravitational fractionation.





Knowing the fractionation coefficients, we retrieve the temperature gradient (noted $\Delta T$) of the firn at different times. The
gravitational effect on isotopes (noted $\delta_{grav}$) is mass dependent, while the thermal fractionation can be approximated as a linear
function of the temperature gradient in the firn, with different fractionation coefficients for nitrogen ($\Omega_{15}$) and argon ($\Omega_{40}$)
gases (Severinghaus et al., 2001). Gravitational and thermal effects on fractionation can thus be disentangled using two pair
of isotopes, as described by the following system:

$$
\begin{cases}
\delta^{15}N = \delta_{grav} + \Omega_{15} \cdot \Delta T \\
\delta^{40}Ar = 4 \cdot \delta_{grav} + \Omega_{40} \cdot \Delta T
\end{cases}
\tag{2}
$$


So $\delta_{grav}$ and $\Delta T$ can be written as a function of $\delta^{15}N$ and $\delta^{40}Ar$:

$$
\begin{cases}
\Delta T = \dfrac{\delta^{15}N - \frac{1}{4} \cdot \delta^{40}Ar}{\Omega_{15} - \frac{1}{4} \cdot \Omega_{40}} = \dfrac{^{15}N_{excess}}{\Omega_{15} - \frac{1}{4} \cdot \Omega_{40}} \\[3mm]
\delta_{grav} = \dfrac{\Omega_{15} \cdot \delta^{40}Ar - \Omega_{40} \cdot \delta^{15}N}{4\Omega_{15} - \Omega_{40}}
\end{cases}
\tag{3}
$$

We use the Eq. (3) to estimate the temperature gradient and gravitational fractionation from the gases isotopes. We use values
of 0.0143 ‰ °C$^{-1}$ for $\Omega_{15}$ and for 0.0386 ‰ °C$^{-1}$ for $\Omega_{40}$ at 230.5 K or –43.7 °C (Grachev and Severinghaus, 2003). The
diffusive column height is calculated from gravitational fractionation using the following equation (simplified from Sowers
et al., 1992):

$$
z_{diffusive} = \frac{R \cdot T}{g} \cdot \delta_{grav}
\tag{4}
$$

where R is the gas constant, T is the temperature, and g is the gravitational acceleration. We used a constant temperature of
229.5 K (–43.7°C), derived from the borehole temperature measurement at 20 m depth in the ABN firn, where season
variations do not reach. The average uncertainty for the $\delta_{grav}$ is 0.008 ‰, resulting in a 1.5 m uncertainty in the diffusive column
height ($z_{diffusive}$). Pooled standard deviation of $^{15}N_{excess}$ in the ice samples give an analytical uncertainty of 0.0027 ‰, resulting
in a $\Delta T$ uncertainty of 0.3°C. Sample preparation includes release of gases in empty lines, and can induce a pressure gradient
leading to small fractionation in both $\delta^{15}N$ and $\delta^{40}Ar$, which is apparently mass-dependant (Severinghaus and Battle, 2006).
In the definition of $^{15}N_{excess}$, mass-dependent processes are cancelled out, and therefore the difference in $^{15}N_{excess}$ between the
two samples is smaller than $\delta^{15}N$ or $\delta^{40}Ar$ alone (0.0137 ‰ for $\delta^{40}Ar$ and 0.0036 ‰ for $\delta^{15}N$).

The temperature difference ($\Delta T$) reconstructed from gases is representative of the gradient between the surface and the lock-
in depth. The lock-in depth can be deeper than the diffusive column height in case of near-surface convective mixing.
Therefore, we also estimate the lock-in depth with the gas-ice depth difference at the same age, from the age models (Sect.
2.3, Fig. 4). The gas-ice depth difference decreases over time because of the compaction in the firn and ice column, so we
deconvolve the depth difference using a mass-conservative compaction model fitted to the modern profile of snow and ice
density measured in the ABN1314 ice core:



$$\int_{z_{lid}}^{z_{surface}} \rho(z)dz = constant = \int_{z_{gas}}^{z_{ice}} \rho(z)dz \tag{5}$$

In short, we compute the mass of ice between the depth of gases and ice of the same age, and find the modern depth where the overlying total ice mass up to the surface is identical. In this model, we assume that the density profile remains the same over time. This estimation of the past lock-in depth is called the deconvoluted $\Delta depth_{ice\text{-}gas}$. Given the ~20 years uncertainty on the age difference, the deconvoluted $\Delta depth_{ice\text{-}gas}$ has an uncertainty of $\pm 3.8$ m.

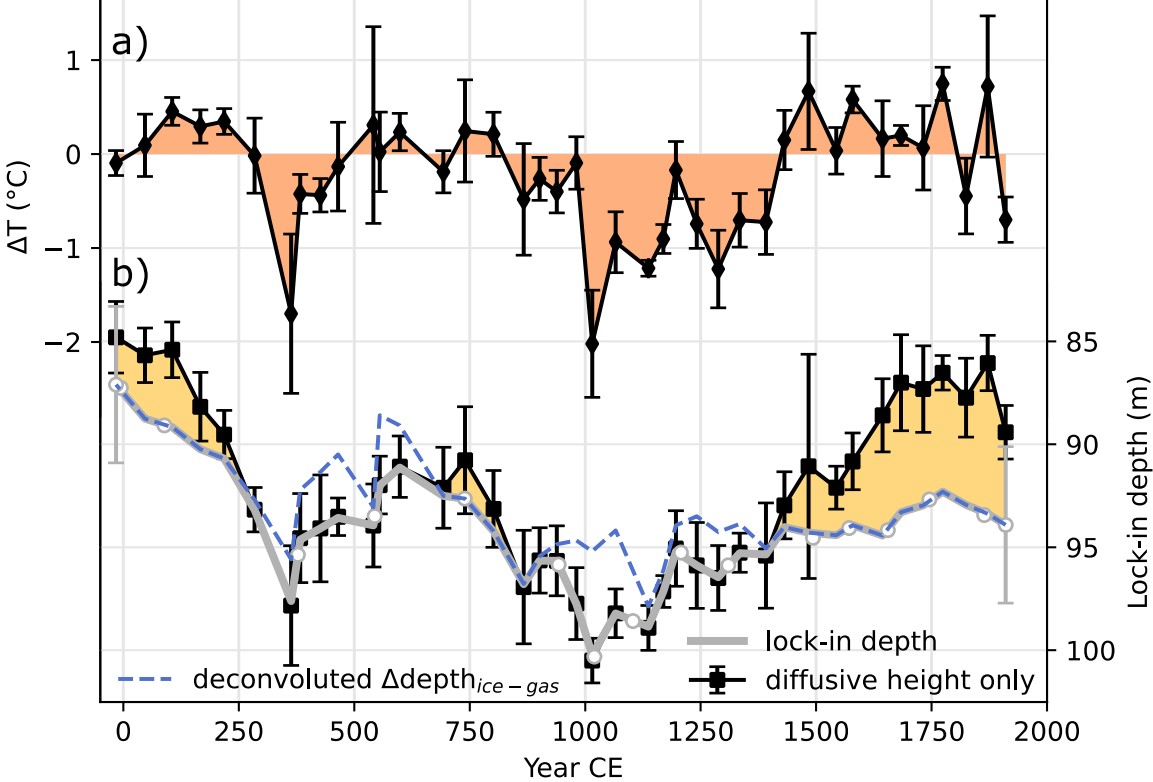

**Figure 7. (a) Series of $\Delta T$ computed from $^{15}N_{excess}$. Orange shadings indicate a warming ($\Delta T>0$) and blue shadings a cooling ($\Delta T<0$). (b) Past lock-in depth (thick grey line) estimated from diffusive column height of gases isotopes (black line with error bars) and gas-ice depth difference (blue dashed line). Yellow shadings highlight the potential presence of a convection zone that would be located in the uppermost layer of the firn (0~5 m depth), when the lock-in depth appears to be deeper than the diffusive column height. For clarity, uncertainties on the lock-in depth are only shown at both ends of the record. White dots on the lock-in depth indicate the ages where the gas age model was tied to WD2014, indicating the constraints on the $\Delta depth$.**

Evolutions of the $\Delta T$, deconvoluted $\Delta depth_{ice\text{-}gas}$ and diffusive column height are shown in Fig. 7. We define the true lock-in depth (Fig. 7b) as the deepest of diffusive column height and deconvoluted $\Delta depth_{ice\text{-}gas}$ because the lock-in depth can be deeper





than diffusive column height in presence of a convective zone, but the lock-in depth should be at least as deep as the diffusive column height. The points where deconvoluted $\Delta depth_{ice-gas}$ (dashed line) is shallower than the diffusive column height are within the 3.8 m uncertainty. They may result from errors in the age model, or in the compaction model: a less-dense than

modern profile would yield greater ice-gas depth differences in better agreement with the diffusive column height. However, improved precision is not required in this study, as errors of ± 3.8 m in the lock-in depth result in ± 0.03°C in the surface to lock-in depth temperature gradient, ten times smaller than the uncertainties on the $\Delta T$ estimated from the gases isotopes. Changes in the firn depth (lock-in depth) are most likely driven by changes in accumulation, which is itself strongly dependant on ice flow and changes in the local slope (Akers et al., 2022; Fig. A3). The main forcing for the lock-in depth is thus not a

temperature factor, we will simply use the value to compute the temperature gradient in the ice sheet ($\Delta T$ between surface and lock-in depth).

### 3.2.2 Firn temperature diffusion modelling and inversion

We use an ice advection and densification model with temperature diffusion in the ice column to simulate the evolution of the firn under different temperature scenarios. This model has been previously described in (Orsi et al., 2012) and we use the

parameters described in the Appendix A4. Briefly, this model uses a prescribed series of surface temperature, snow accumulation, and bottom geothermal flux to simulate the evolution of temperature in the firn and ice column. In our simulations, densification follows the modern density measurements, and the lock-in depth is inferred from the depth determined previously with deconvoluted $\Delta depth_{ice-gas}$ and diffusive column height. We create 220 temperature simulations by adding a small perturbation ramping up to +1°C on 10 years to a base temperature history estimated from Dome C (Jouzel et

al., 2007) and ice-flow related temperature changes (Mouginot et al., 2019, details of the temperature forcings in Fig. A14 and Appendix A4).

We find the optimized temperature history that fits both the present-day borehole temperature profile and the past $\Delta T$ temperature gradients estimated from the $^{15}N_{excess}$. The optimized temperature history is estimated by least square regression of a linear combination of the 220 temperature simulations to the borehole temperature profile (e.g., Orsi et al., 2012) and the

temperature gradients estimated with $^{15}N_{excess}$ (e.g., Orsi et al., 2014). In this study, we use both borehole and gas data to constrain the temperature history; the borehole temperature profile constrains the long-term changes while the $^{15}N_{excess}$ constrains temperature changes at the scale of ~20 to ~200 years (the diffusion time in the firn column of gases and temperature, respectively). However, the two datasets result in a mismatch on the long-term trends, with $^{15}N_{excess}$ suggesting a cooling trend. To reconcile the $^{15}N_{excess}$ with the borehole temperature profile, a correction of + 0.0046 ‰ is applied on $^{15}N_{excess}$: this

minimizes the standard deviation of the temperature reconstruction as it tries to squeeze in rapid temperature changes to arrange for diverging datasets if $^{15}N_{excess}$ is left uncorrected (Figs. A16 and A17). We hypothesize that this $^{15}N_{excess}$ correction for ice samples could be related to expulsion of gases through ice matrix during bubble formation in the firn to ice transition (Severinghaus and Battle, 2006), although the effect on $^{15}N_{excess}$ has never been clearly quantified. We apply the same





correction on all points because it results from physical processes during the formation of bubbles, and we consider it to remain

constant at the timescales studied here. A similar correction of + 0.0037 ‰ has been used at WAIS Divide (Orsi, 2013), based on the mismatch between firn air and shallow ice measurements. This correction implies that the long-term trend is constrained by borehole temperature rather than $^{15}N_{excess}$ absolute values. Further work in the firn to ice transition is needed to better quantify the effect of bubble formation on gases isotopic composition.

Ice flow is expected to affect the temperature profile in the ice at ABN, because it advects ice from a location further upstream

that was colder when the snow deposited. This can explain the presence of cold ice at depth under ABN, which could be misinterpreted by the inversion as a climatic warming trend. To account for this effect, we consider the ice in the diffusion-advection model in a Lagrangian perspective and dissociate temperature changes caused by site displacement and climatic temperature changes (details and justifications are given in Appendix A4). After inversion of the model, the ice-flow related temperature changes is subtracted in order to retrieve the climatic temperature changes (Fig. 8). This relies on the assumption

that ice flow is in a steady state and the spatial temperature gradient remained constant over the past 2000 years during which we present our temperature reconstruction.

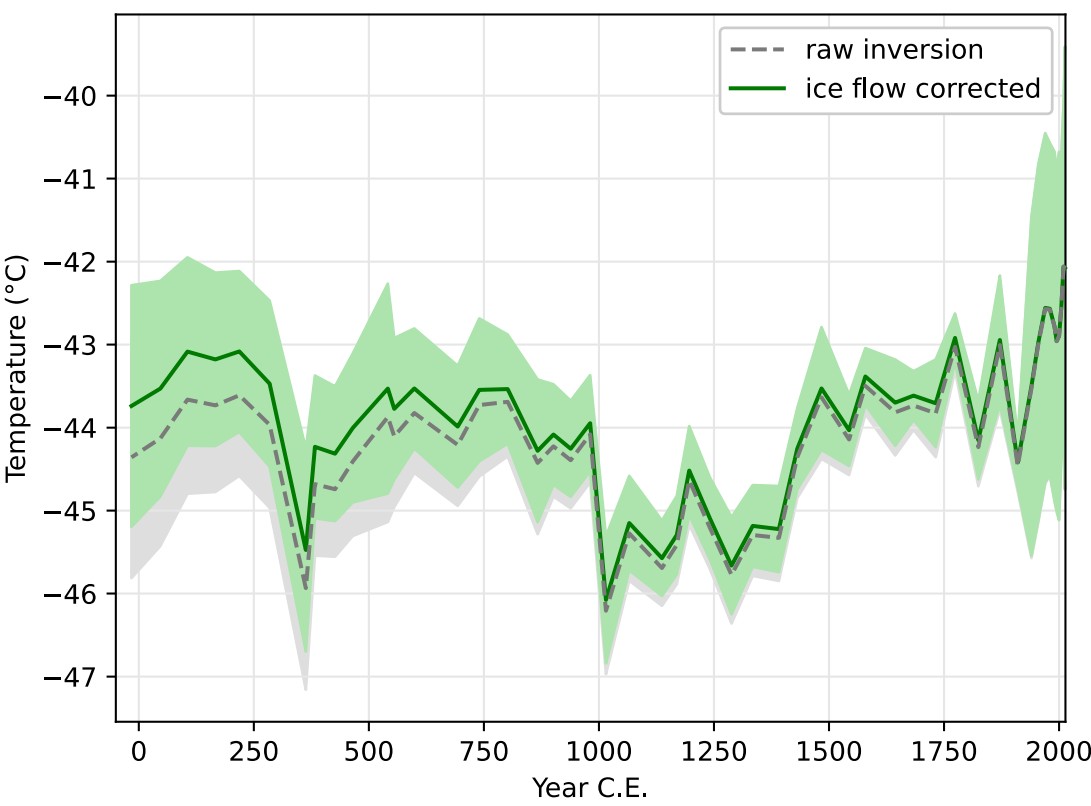

**Figure 8. Temperature history reconstructed from the $^{15}N_{excess}$ and borehole temperature inversion. Black dashed line shows the raw measurement and green line shows the glaciology-corrected values. The shading represents errors estimated from the inversion, which depend on measurement precision.**

Uncertainties of the diffusion model inversion are calculated following the method described by Orsi et al. (2014), where uncertainty of model parameters are scaled to the uncertainty on $\Delta T$ and borehole temperature data used for the inversion. A smoothing parameter is used to constrain the inversion, by limiting the degree of variability of two temperature simulations with perturbation occurring close together in the linear combination used for the inversion. We set the inversion to use an exponentially decreasing covariance in the linear combination, which reaches 0.5 for a time difference of 70 year, roughly twice the time resolution of the gas constraints on $\Delta T$. Finally, a signal to noise ratio parameter is adjusted to force the inversion temperature to fit to the real borehole and $\Delta T$ data points within the error, with a minimum cost on the inversion uncertainty (Figs. A18 and A19). The average uncertainty on the temperature reconstruction is $\pm 0.7°C$.



## 4. Discussion

### 4.1 Differences between the reconstructions

The temperature at the ABN ice core site is reconstructed from water stable isotopes (hereafter called $\delta^{18}O$ temperature), and from the inversion of the temperature profile in the borehole and past temperature gradients estimated from inert gas stable isotopes (hereafter called $^{15}N_{excess}$ temperature, Fig. 9). The $\delta^{18}O$ temperature is relatively constant, supporting stable conditions of about −42°C, with less than 1°C change during the past 2700 years, whereas the $^{15}N_{excess}$ temperature is marked by changes of temperature of an amplitude of 2 to 3°C, with cold periods from 300 to 450 CE and from 1000 to 1400 CE, and a recent warming of about 1°C. Although the temperature reconstructions use measurements from the same core, the material used for the reconstructions differ fundamentally, which can explain some of the disparity.





**Figure 9.** Comparison of (a) δ¹⁸O temperature and ¹⁵N$_{excess}$ temperature reconstructions with (b) upstream elevation and (c) upstream slopes and (d) d$_{ln}$ in the ABN1314 ice core. Distance upstream is indicated on the top axis, and time is indicated on the bottom axis, the correspondence is relying on the assumption that the ice flowed at a constant rate of 15.4 m yr⁻¹ (41.5 km in 2700 years). Error shades in (a) are the error depending on slop used for the reconstruction from δ¹⁸O, and the same as in Fig. 8 for ¹⁵N$_{excess}$.

First, regarding the time significance, water isotopes in ice are accumulated intermittently during precipitation events, while

gases are continuously available, and the temperature permanently diffuses in the ice. The sporadic nature of precipitation and

485

490



the consistent warm anomaly associated with precipitation at ABN may cause water isotopes to inaccurately represent the mean temperature (Servettaz et al., 2020). In particular, the cold and dry conditions that can occur on the East Antarctic Plateau are likely to be under-represented in the water isotopes, because a large part of the snow accumulation can be attributed to a few precipitation events (Turner et al., 2019). This may cause the $\delta^{18}O$ temperature to misrepresent the on-average colder periods if they are still interrupted by less-frequent but similarly warm precipitation events, or if the seasonality of precipitation changes with a relatively increased contribution in summer during these cold periods. Additionally, long periods without snow precipitation contribute to increase the snow $\delta^{18}O$ by preferential sublimation of light isotopes (Hughes 2021), smoothing out the signature of cold periods. Here, the precipitation events continue to carry the water isotopes with a constant signature of −42°C. On the other hand, due to the slow diffusion of gases in the firn, the signal integrates temperature variation over a time window of a few decades (Witrant et al., 2012; discussed in Appendix A5), but effectively records the temperature even if there are changes in the accumulation regime. Therefore, changes in the seasonality of precipitations with a shift to drier winters could explain both colder conditions in the winter and a lack of snow accumulation, resulting in the failure of water isotopes to capture the cold conditions in the record (Servettaz et al., 2020).

Second, there is a spatial discrepancy between the two reconstructions: the $\delta^{18}O$ temperature signal is initially acquired in the atmosphere, during the condensation of the precipitation when condensate phase is exchanging with water vapour (Jouzel and Merlivat, 1984), even if it can be modified later (Münch et al., 2017; Casado et al., 2018). The ice temperature is constrained by the snow surface temperature, and the gases in the firn are at equilibrium with the surrounding ice, which temperature thus controls the gases thermal fractionation. Consequently, temperature changes in the snow could be exacerbated by changes in the inversion layer conditions: a more stable inversion layer causes the snow surface temperature to cool much more than the 2 m temperature that was used to calibrate the $\delta^{18}O$ – temperature slope. If changes in the temperature inversion were caused by atmospheric temperature changes, we would still expect the $\delta^{18}O$-derived temperature reconstruction to reflect the changes observed in the gas-based temperature reconstruction, albeit with a weaker amplitude. This has been the case for example for glacial – interglacial transition in West Antarctica: the snow temperature has varied with greater amplitude than what the water isotopes have recorded (Buizert et al., 2021). However, here, the cold periods identified with the gas and borehole temperature reconstruction are not matched by any sign of cooling in the $\delta^{18}O$ record, suggesting that the differences are not entirely caused by atmospheric temperature changes amplified by the temperature inversion.

Another possible origin for the difference between atmospheric and surface snow temperatures is that surface winds weaken the inversion by causing turbulent mixing of the stratified air layers near the surface (Hudson and Brandt, 2005; Pietroni et al., 2014). Given that the average katabatic wind speed increases with the terrain slope (Parish and Waight, 1987; Parish and Bromwich, 1991; Vihma et al., 2011), we investigated possible effects of elevation and topographic slope changes on the recorded temperature (Fig. 9). Although it was taken along the ice stream flowline and not the katabatic wind flowline, the terrain elevation and slope roughly provides information of what slope-induced changes in katabatic winds may have affected the ABN site as it drifted (Fig. 9b and 9c). It appears that ABN site was on a plateau with a null slope between 18 and 8 km

 

upstream from the coring location, which roughly corresponds to the 800 to 1500 CE period, slightly longer than the cold

period identified with $^{15}N_{excess}$ temperature (Fig. 9a). It is possible that a reduction of katabatic winds on this part of the terrain

allowed for a stronger near-surface temperature inversion to settle, effectively cooling the snow relative to the atmosphere.

Nevertheless, the recent warming of 1°C cannot be attributed to changes in the slope as the warming is occurring while the

slope gets gentler.

Influence of temperature on $\delta^{18}O$ could be masked by a concurrent change in moisture source: e.g., if cold periods were

associated with a shift towards warmer source, the two effects could compensate each other and result in a constant $\delta^{18}O$. Such

a change in moisture source can be recorded in the $d_{ln}$, which efficiently tracks changes in the moisture source temperature

(Uemura et al., 2012; Markle and Steig, 2022). The $d_{ln}$ series shows no significant trend, with the values averaging 13.6 ‰.

When averaged on 30-years to smooth out noise, the standard deviation of $d_{ln}$ is only 0.6 ‰, with no remarkable change during

the cold periods in the $^{15}N_{excess}$ temperature. This suggests that there was minimal to no change of the moisture source affecting

the variability of $\delta^{18}O$, and thus confirms that the condensation temperature remained relatively stable for the last 2700 years.

In a recent study, Morgan et al. (2022) suggest that the gas stable isotopes in the firn could be affected by seasonal rectification:

in absence of mixing of air in the surface layer, the winter temperature inversion cools the snow surface and densifies the near-

surface firn air which could sink and advect the air column downward more efficiently than during summer. Winter advection

of air down into the firn lowers the $^{15}N_{excess}$ isotopic signal, which can result in an apparent colder $\Delta T$. Sinking air is suspected

to occur when katabatic wind and surface turbulence are weak, which allow a strong temperature inversion to develop. On the

other hand, strong katabatic winds induce a mixing of both the air above the snow surface where temperature inversion is the

strongest, and the air in the uppermost layer of the firn, increasing the convection layer. Morgan et al. (2022) hypothesize that

the change in surface slope and curvature, which affect the strength of katabatic winds (Vihma et al., 2011), may in turn be

responsible for some difference in the $\Delta T$ derived in the isotopes, along with snow accumulation and the lock-in depth. Surface

topography changes are linked to the glacial flow both at South Pole (Morgan et al., 2022) and ABN, two Antarctic sites that

are not on an ice divide.

At ABN, the periods with suspected upper firn convection (yellow shadings, Fig. 7b) correspond to periods with positive $\Delta T$

(orange shading, Fig. 7a), whereas periods with deepest lock-in depths are associated with very negative $\Delta T$. The existence of

a convective zone may be linked to the surface wind speed, as ABN was in the steeper part of the slope during the periods with

a convective zone (Fig. 9b). However, the late Holocene conditions are unlikely to result in a strong rectifier effect at ABN,

because this site is located on a slope where there is expected sustained surface winds, and even at South Pole where

temperature is on average 7°C colder than ABN does not support a rectifier effect on the Holocene (Morgan et al., 2022).

Although the $\Delta T$ – lock-in depth correlation may be linked to past variations of winds caused by topographic slope, low

temperatures resulting from climate variability may also be responsible for an increased lock-in depth due to slower

densification (Goujon et al., 2003). Rather than a firn rectifier effect, the ABN temperature reconstruction may be simply





reflecting changes in the snow temperature mirroring near-surface temperature inversion strength, influenced by average wind speed and local slope.

To summarize, there is possibility that water isotopes are biased towards warm temperatures because of lack of precipitation in cold periods, and that gas isotopes reflect topography-driven changes in wind speed and temperature inversion strength. We expect the temperature history to be in between the two estimations, although the $^{15}N_{excess}$ should more consistently record temperature changes at the snow surface.

### 4.2 History of temperature across the East Antarctic Plateau

Many regions of the globe including Antarctica have been marked by a cooling trend during the past 2000 years, until the beginning of the industrial era (PAGES 2k Consortium, 2013). Climate history of the Antarctic region was reconstructed from ice core water isotopes (Stenni et al., 2017), with some cores on the East Antarctic Plateau where ABN is located (EDC, TALDICE), and the coastal region of Wilkes land (Law Dome, location of coring sites is indicated on Fig. 1). We selected ice cores from the Antarctica2k database (Stenni et al., 2017) that cover at least the past millennium until the pre-industrial era

(1000–1900 CE) and plotted the $\delta^{18}O$ or $\delta D$, representative of the temperature, and their 1000 to 1900 CE trend for each core (Fig 10). To avoid biases based on differing calibration methods, comparisons are made using isotope values directly ($\delta D$ or $\delta^{18}O$). For comparison, each core was resampled at a 30-year resolution, and the trends were computed on the 900 years spanning from 1000 to 1900 CE (31 points); trends are significant if $r^2 \geq 0.13$ (p-value < 0.05). While Stenni et al. (2017) report a general cooling trend both for Wilkes coast and East Antarctic Plateau on the past 2000 years, we find that the last

millennium is not marked by any significant trend in the plateau ice cores, when taken separately. On the other hand, the coastal ice core from Law Dome shows a slight cooling trend (decreasing $\delta^{18}O$). The $\delta^{18}O$ record from the ABN1314 ice core has a higher temporal resolution than other East Antarctic Plateau ice cores, but its absence of a significant trend supports the previous findings pointing to stable atmospheric conditions during the past 2000 years on the East Antarctic Plateau. However, the $\delta^{18}O$ records on the east Antarctic Plateau are subject to debate whether they can effectively reflect rapid temperature

changes, due to low accumulation rates and intermittent accumulation events (Münch et al., 2017; Casado et al., 2020). In any case, the $^{15}N_{excess}$ temperature variability strongly contrasts with $\delta^{18}O$ records from ABN1314 and other East Antarctic Plateau ice cores.






**Figure 10. Comparison of δ¹⁸O records from ABN1314 core and a selection of nearby ice cores from the Antarctica2k database (Stenni et al., 2017). Thin lines: full resolution, thick lines: resampled at 30-year resolution. Trends are computed from the 30-year resampled series, and are significant if r²>0.13 (p-value < 0.05, n = 31)**


In particular, in the $^{15}N_{excess}$ and borehole-based reconstruction (Fig. 11a), the temperature at ABN varies between –44.5°C and –43°C after 1750 CE but stabilizes at an average of –42.5°C after 1975 CE. After 1900 CE, the temperature inversion is mainly constrained by the borehole temperature, and this warmer phase can be seen in the steepening gradient of the temperature above 100 m below surface (Fig. 3). This temperature of –42.5°C is about 1°C warmer than the 1500–1850 CE

average and could reveal the effect of recent warming in East Antarctica. This surface warming at ABN is unlikely to be caused





by a topographic change as the slope is flattening near the drilling site (Fig. 9) and would on the contrary favour the slowing of katabatic winds and surface cooling by strengthening of the near-surface temperature inversion. The absence of further warming on the East Antarctic Plateau after 1975 is consistent with observations and could be related to the compensation effect associated with the positive trend in the southern annular mode in the later part of the 20th century (Nicolas and

Bromwich, 2014; Fogt and Marshall, 2020). A similar warming of about 1°C inferred from East Antarctic borehole temperatures has been reported in locations near the ice divide of Dronning Maud Land, but was equivocal, as a borehole off the divide showed a possible cooling trend except for the most recent couple of years (Muto et al., 2011). The $^{15}N_{excess}$ and borehole temperature reconstruction provides new insight on the climate of East Antarctica that may complement the many $\delta^{18}O$ records in this region. Three independent sources of data support this varying temperature history: the borehole

temperature, the gas $^{15}N_{excess}$ and the lock-in depth. Together they consolidate the evidence that surface temperature changed with a greater amplitude than what $\delta^{18}O$ suggests.

## 4.3 Atmospheric teleconnections

In Antarctica and the Southern Hemisphere, the main mode of atmospheric variability is the Southern Annular Mode (SAM), with a strong signature on Antarctic temperature (Marshall and Thompson, 2016). The SAM describes the surface pressure

difference between mid and high latitudes (Marshall, 2003; Fogt et al., 2009), and a positive phase denotes a strong pressure gradient. On the East Antarctic Plateau, SAM phase and surface temperature are anti-correlated because a positive SAM phase is associated with a reduced meridional heat transport (Marshall and Thompson, 2016), and the SAM signature is found in the temperature at the ABN site, although its effect on $\delta^{18}O$ is less prominent (Servettaz et al., 2020). The variability of SAM is linked to other climate modes and parameters such as El Niño – Southern Oscillation (Abram et al., 2014), Atlantic Meridional

Overturning Circulation (Pedro et al., 2018) and solar irradiance (Wright et al., 2022).





**Figure 11. (a)** δ18O temperature and 15Nexcess temperature reconstructions (this study). Error shades are the same as in Fig. 9. **(b)** Southern Annual Mode (SAM) annual reconstruction (Dätwyler et al., 2018). Thin lines show the annual reconstruction. **(c)** West Antarctic Ice Sheet (WAIS) Divide ice core δ18O (Steig et al., 2013). Thin lines show the 5-year average. **(d)** Taylor Dome δ18O (Steig et al., 2000). Thin lines show the full resolution. All thick lines in this Figure Ahow the 30-year average, except for 15Nexcess temperature which has a resolution of about 45 years. Yellow shading highlights the 1000–1400 CE period during which the 15Nexcess temperature is significantly colder, in phase with a positive SAM index. WAIS Divide and Taylor dome δ18O data were taken from the Antarctica2k database (Stenni et al., 2017), where Taylor dome is on its updated TD2015 timescale (Sigl et al., 2014).





On the time scale of a thousand years, the SAM has been reconstructed from paleoclimate proxies sensitive to SAM-related temperature anomalies (Abram et al., 2014; Dätwyler et al., 2018). The two reconstructions yielded similar trends, although the reconstruction of Abram et al. (2014) shows more amplitude due to being calibrated on the instrumental period with annual SAM indices (Wright et al., 2022). Here we will only show the annual SAM reconstruction from (Dätwyler et al., 2018; Fig. 11b), which relies largely on the same proxies, but with a correlation plus stationarity criterion for proxy selection. During the 1000–1400 CE period, the SAM was in a relatively positive phase, followed by a net switch to a strongly negative phase that lasted until about 1800 CE. The $^{15}N_{excess}$ temperature at ABN is remarkably consistent with the SAM variability, as it shows colder temperatures during the positive phase of the SAM in the beginning of the last millennium, with rapidly increasing temperature as the SAM changes to the negative phase. Although the SAM reconstructions do not extend beyond 1000 CE, sea ice proxies in the nearby Adélie Basin (66° S, 140° E) suggest that the positive SAM phase may have started around 830 CE (Crosta et al., 2021), which is sensibly earlier than the $^{15}N_{excess}$ temperature decrease observed at ABN at around 1000 CE.

Across Antarctica, the West Antarctic Ice Sheet (WAIS) Divide ice core shows a shift in $\delta^{18}O$ from −33.2 ‰ on the 0–1000 CE period to −33.8 ‰ on the 1000–1400 CE period (Fig. 11c, Steig et al., 2013). Although less abrupt than the change observed in $^{15}N_{excess}$ temperature, this shift may reflect a change driven by positive SAM on West Antarctic temperature. Moreover, particle size distributions in the WAIS Divide ice core suggest that a poleward contraction of Southern Westerly Wind belt during the 1050–1430 CE period, marking a positive SAM phase (Koffman et al., 2014). Nevertheless, the transition to a negative SAM phase after 1400 CE is not accompanied by a warming in West Antarctica, suggesting that other mechanisms compensate for the SAM effect on WAIS Divide ice core $\delta^{18}O$.

By the same way SAM is defined as the first Empirical Orthogonal Function (EOF) of geopotential height, the Pacific South American modes PSA1 and PSA2 are the second and third EOFs of geopotential height, and represent the zonally asymmetric variability in the Pacific sector of Antarctica (Marshall and Thompson, 2016). Even though its intensity is weaker, the PSA2 has a more widespread temperature effect on Antarctic temperatures: a positive PSA2 is associated with warm anomalies in East Antarctica, while cold anomalies affect West Antarctica and the Eastern Ross Sea (Marshall and Thompson, 2016). The opposing trends in the $^{15}N_{excess}$ in ABN, East Antarctica (+2.9°C 1000 yr$^{-1}$, r$^2$ = 0.76) and $\delta^{18}O$ in WAIS Divide Core in West Antarctica (−0.2 ‰ 1000 yr$^{-1}$, r$^2$ = 0.07) are in line with an increase of the PSA2 phase over the 1000–1900 CE period. An increase in the PSA2 phase could indeed explain the cooling trend observed in both in WAIS Divide and Taylor Dome (−0.7 ‰ 1000 yr$^{-1}$, r$^2$ = 0.13; Fig. 11d, Steig et al., 2000), despite the change to a more negative SAM (−0.7 1000 yr$^{-1}$, r$^2$ = 0.19). The low significance of trends in WAIS Divide core $\delta^{18}O$ could result from the compensating effects of a switch to a more positive PSA2 (cooling effect on WAIS) and a more negative SAM (warming effect on WAIS).

Increased katabatic winds in the Eastern Ross Sea under a transition to a positive phase of PSA2 could support an opening of the polynya in the East Ross Sea around that time, as suggested by an increase of $\delta D$ in the Roosevelt Island Ice Core in the Ross Sea (Marshall and Thompson, 2016; Bertler et al., 2018). The decreasing accumulation of both Roosevelt Island and West Antarctic Ice Sheet during the last millennium (Bertler et al., 2018) is also compatible with a positive trend in the PSA2





(Marshall et al., 2017). Such a change in the PSA2 would have atmospheric and oceanic implications beyond the few records discussed here, and it is still unclear what is the long-term trend of secondary climate modes in Antarctica, such as PSA1 or PSA2. Despite their relatively low contribution to climate variability on instrumental era, it is possible that the long-term

variability of PSA modes had a greater amplitude owing to changing atmospheric conditions in the Ross Sea related to the ongoing retreat of the Ross ice shelf during the late stages of the Holocene (Yokoyama et al., 2016). The increasing number of high-resolution paleoclimate records in the Antarctic region may help to investigate the past variability of secondary atmospheric modes, similarly to what has been done for the SAM (Abram et al., 2014; Dätwyler et al., 2018), but this is beyond the scope of this study.

**5. Conclusion**

We have reconstructed past temperature variability over the last 2000 years from an ice core drilled at Aurora Basin North (ABN), inland East Antarctica. We used two temperature reconstructions: (1) based on water stable isotopes ($\delta^{18}$O) and (2) by inversion of a diffusion model matching borehole temperature and past temperature gradients recorded in gases stable isotopes ($^{15}$N$_{excess}$). The ABN drilling site is located far from a divide, so we carefully took the ice flow into account when estimating

the past temperature, and we were able discuss the climate variability in East Antarctica at a high resolution on the past 2000 years.

The two temperature reconstructions from the same ice core show major differences, that are attributed to the difference of spatial and temporal significance of the proxies used: the water stable isotopes acquire their signal during condensation in the atmosphere and are accumulated sporadically during snowfall events; whereas the borehole temperature and gases are

constantly exchanging with snow surface conditions, and thus integrate the snow surface temperature over a few decades. While the $\delta^{18}$O temperature reconstruction shows little to no trend over the past 2000 years, consistently with other East Antarctic Plateau ice cores, the $^{15}$N$_{excess}$ and borehole temperature inversion suggests that surface temperature varied with an amplitude of about 2°C. A cold period during 1000–1400 CE matches a positive phase of the Southern Annular Mode (SAM), known to have a cooling effect on most of Antarctica (Marshall and Thompson, 2016). The surface warming at ABN after

1400 CE contrasts with West Antarctic $\delta^{18}$O records and indicates the influence of zonally asymmetric Pacific-South American atmospheric modes. Prevalent cold and dry winter during cold periods may explain both the inability of water isotopes to record temperature changes, and the lower-than-average surface snow temperature. The $^{15}$N$_{excess}$ temperature reconstruction seems to better track surface temperature changes than $\delta^{18}$O, albeit with a lower temporal resolution, as shown by its remarkable consistency with SAM variability. This work highlights the importance of using diverse proxies in temperature

reconstructions, and motivates further studies to understand the proxies used, beyond simple linear relationships.





# Appendix A: Supplementary information on data production

## Appendix A1 Age Models

**Figure A1. Age model for the ABN1314 core. The grey shading indicates cumulated uncertainties on annual layer counting. Volcanic**
**horizons were identified on the sulphur record and were tied to the ages of volcanic events on WD2014 timescale (Sigl et al., 2015,**
**2016), and given a 5-year uncertainty, corresponding to the maximum intrinsic uncertainties on WD2014 timescale for this period.**







**Figure A2. Methane concentration in trapped air in ABN1314 (blue) averaged on 1 m sections (the light blue shading shows the 1σ standard deviation of measurements in the 1 m section, measured on Continuous Flow Analysis system), West Antarctic Ice Sheet** 695 **(WAIS) Divide (red, Mitchell et al., 2011), and Law Dome (orange, Rubino et al., 2019), on WD2014 gas timescale (Sigl et al., 2015, 2016). Vertical axis changes scale with an axis break at 750 ppb.**



**Appendix A2 Ice-flow related changes**



**Figure A3. Upstream elevation, slope, curvature, and accumulation compared to the ice accumulation in the ABN1314 core. ABN**
**ice accumulation was calculated with the layer thickness from Annual Layer Counting and the density profile measured by weighing**
**known volumes of the ice core, and is shown with a 20-year running mean to smooth out the reconstruction. The light blue shading**
**shows the 20-year standard deviation. Upstream accumulation is estimated by integrating the ice mass between surface and first**
**isochrone reflector in the ground penetrating radar profile. We used the density profile measured in the ABN1314 ice core to estimate**
**the mass, assuming the density upstream follows a similar profile, and we inferred the age of the isochrone from the age at the**
**corresponding layer in the ABN1314 core. Ice movement since deposition is estimated by varying the spatial length of the upstream**
**accumulation record sampled to maximise the correlation with the 2700 year temporal accumulation record from the ABN ice core**
**(lower panel).**





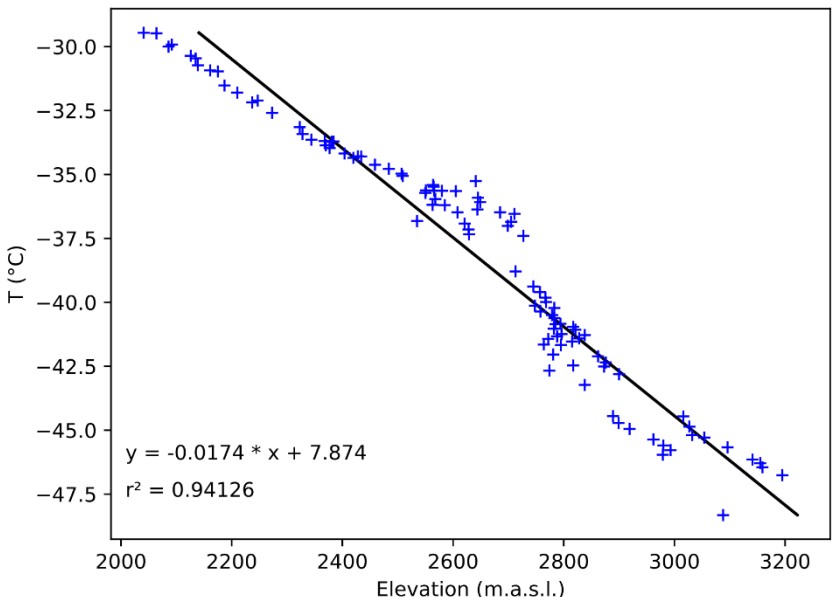

**Figure A4. Temperature – elevation slope in the 2000 – 3250 m.a.s.l. elevation range in East Antarctic traverses with site temperature estimations (Xiao et al., 2013; Pang et al., 2015).**



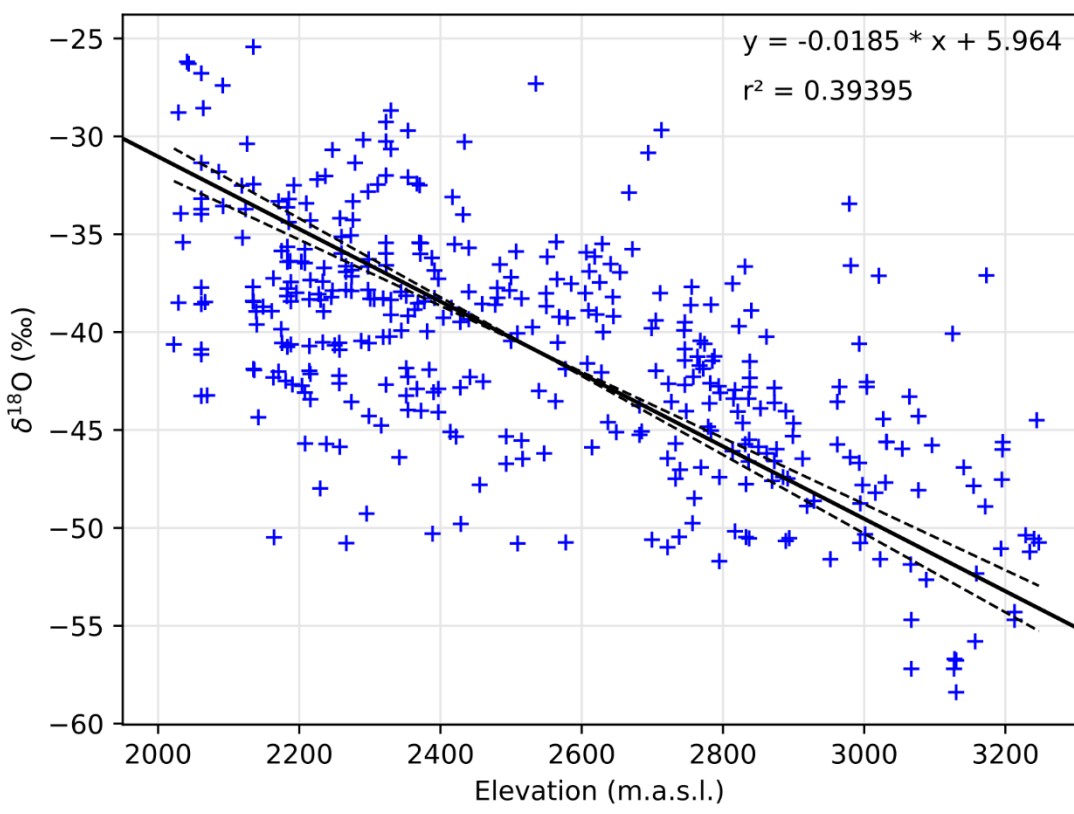

**Figure A5. δ¹⁸O – elevation slopes in surface snow studies: Zongshan to Dome A traverses (Xiao et al., 2013; Pang et al., 2015) and (Goursaud et al., 2018). We restricted our slope to sites at elevations comprised between 2000 and 3250 m.a.s.l., and longitudes from 80°E to 160°E.**





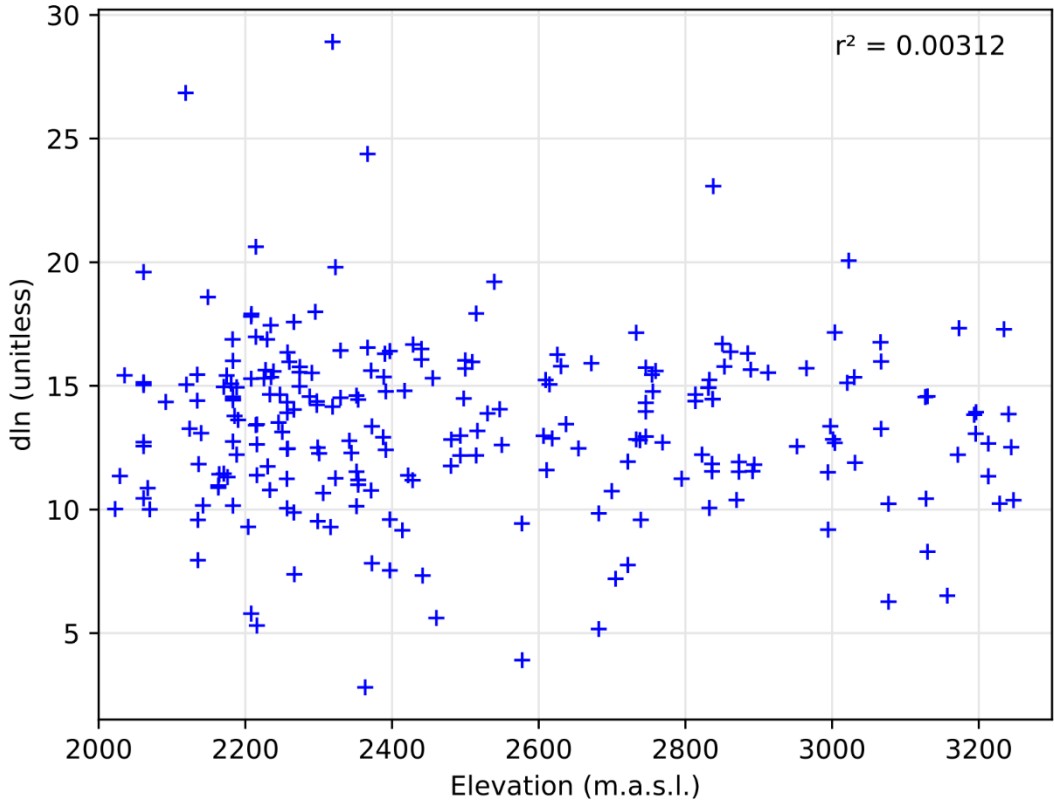

**Figure A6. $d_{ln}$ – elevation scatterplot in (Goursaud et al., 2018) database, restricted to sites at elevations comprised between 2000 and 3250 m.a.s.l., and longitudes from 80°E to 160°E.**

**Appendix A3 Calibration and corrections of isotopic composition of gases**

Measurement of gases from ice cores and their correction have been extensively described by Severinghaus et al. (2001). In addition to their corrections of pressure imbalance (section 3.1 of this document) and chemical slope (section 3.2), we introduce a specific correction for focus drift (section 3.3) before the normalization to atmosphere, which is the standard for the gases isotopes measured in this study (section 3.4).

**Appendix A3.1 Pressure imbalance correction**

In a dual inlet mass spectrometer, two bellows alternatively supply gas to the spectrometer source. Possible differences in the pressure of each bellow can influence the beam intensity and thus measured δ values. This effect quantified using daily-determined Pressure Imbalance Slopes (PIS), where the volume in the bellows is forced into asymmetrical state to evaluate the effect on isotope ratios (for example Fig. A7). The pressure imbalance (ΔP) is defined by the difference of intensity (*int*) between sample and standard of the main gas measured ($^{28}N_2$ in nitrogen configuration or $^{40}Ar$ in argon configuration).





$$PIS_{{}^{40}Ar} = \frac{\Delta\delta^{40}Ar}{\Delta P}$$

(1)

Isotopic measurements are then corrected using daily PIS and imbalance during measurement, as shown in Eq. (2).

$$\delta^{40}Ar_{PI\ corrected} = \delta^{40}Ar_{raw} - PIS_{{}^{40}Ar} \times \Delta P$$

(2)

Equivalent slopes and corrections were performed for $\delta^{15}N$. $\Delta P$ is calculated with intensities averaged on entire blocks for standards (11 integrations) and samples (10 integrations). Pressure imbalance correction is applied block by block, and the standard deviation of 5 blocks of argon in a sequence typically decreases from 0.020 ‰ to 0.015 ‰ (0.005 ‰ improvement).

The pressure imbalance corrected values are compared to raw values in Fig. A8. The Pooled Standard Deviation (PSTD) in Fig. A8 is given for ice replicates, and is therefore larger than the standard deviation of block-averaged δ values within a sequence, for a unique sample.

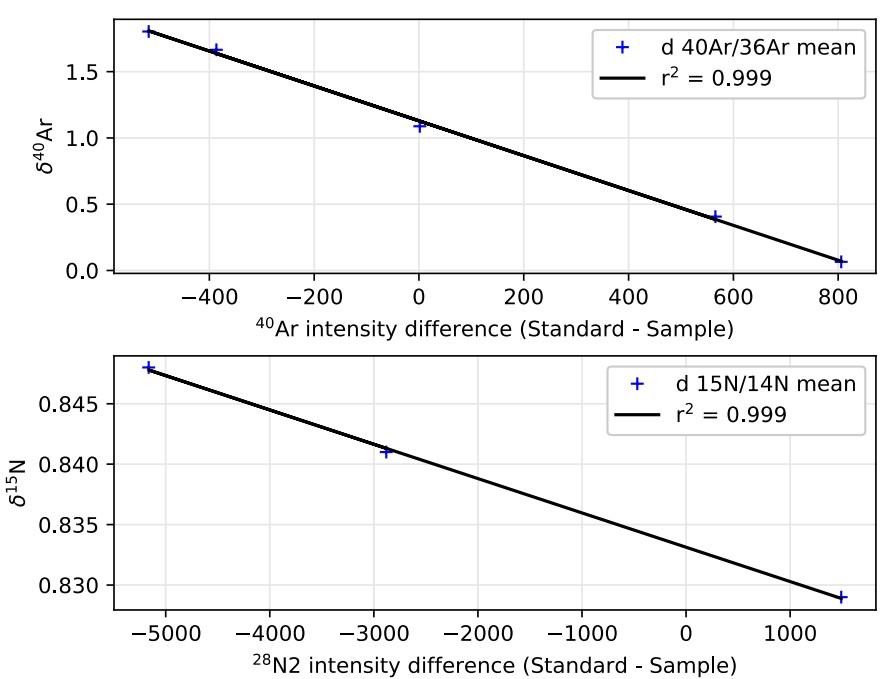

**Figure A7. Variability of $\delta^{40}Ar$ and $\delta^{15}N$ as a function of intensity difference between sample and standard.**





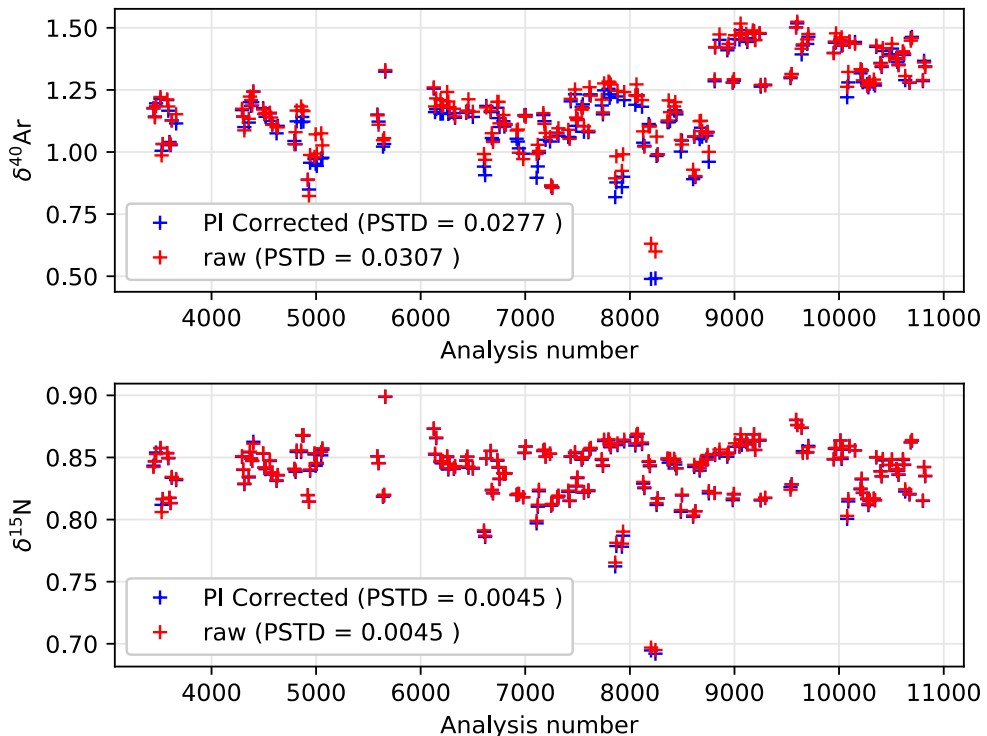

**Figure A8. Comparison of δ⁴⁰Ar and δ¹⁵N before and after pressure imbalance correction for all the measurements. Analysis number corresponds to the number of blocks proceeded, measurements were performed between February and December of 2019. Pooled Standard Deviation (PSTD) of each series are indicated in the captions.**


**Appendix A3.2 Chemical Slope correction**

The gas supplied to the spectrometer and ionized by the filament is a mixture of argon (about 1 %) in nitrogen (about 99 %) with very few other gases. The elemental ratio of argon to nitrogen can modify how effectively the argon will be ionized, because of charge transfer in the source. Therefore, the relative quantity of argon in the mixture can affect the $\delta^{40}$Ar measured by the spectrometer. The chemical ratio effect is quantified by preparing collector tubes of pure nitrogen with varying amount of standard argon of known isotopic composition. We determine the ratio of Ar/N$_2$ with peak jumping, and measure the $\delta^{40}$Ar

over 5 blocks for each mixture. The slope is defined in Eq. 4, and illustrated by Fig. A9.

$$CS_{^{40}Ar} = \frac{\Delta\delta^{40}Ar_{PI\ corrected}}{\Delta\delta Ar/N_2} \tag{3}$$

We apply the chemical slope correction on the $\delta^{40}$Ar values averaged on a full sequence, because the elemental mixture does not change within a sample.

$$\delta^{40}Ar_{CS\ corrected} = \delta^{40}Ar_{PI\ corrected} - CS_{^{40}Ar} \times \delta Ar/N_2 \tag{4}$$

The chemical slope was measured after each change in the filament and if the filament was put in contact with air, as it may cause change in the ionization rate and therefore the chemical slope. Consequently, we only have three chemical slope





measurements for 8 months: at the beginning, in the middle and at the end. This correction does not result in an immediate improvement of the Pooled Standard Deviation (Fig. A10), because the chemical slope error is partly compensated by a drift error (next section), so correcting only the first effect induces a small increase in the sample difference.

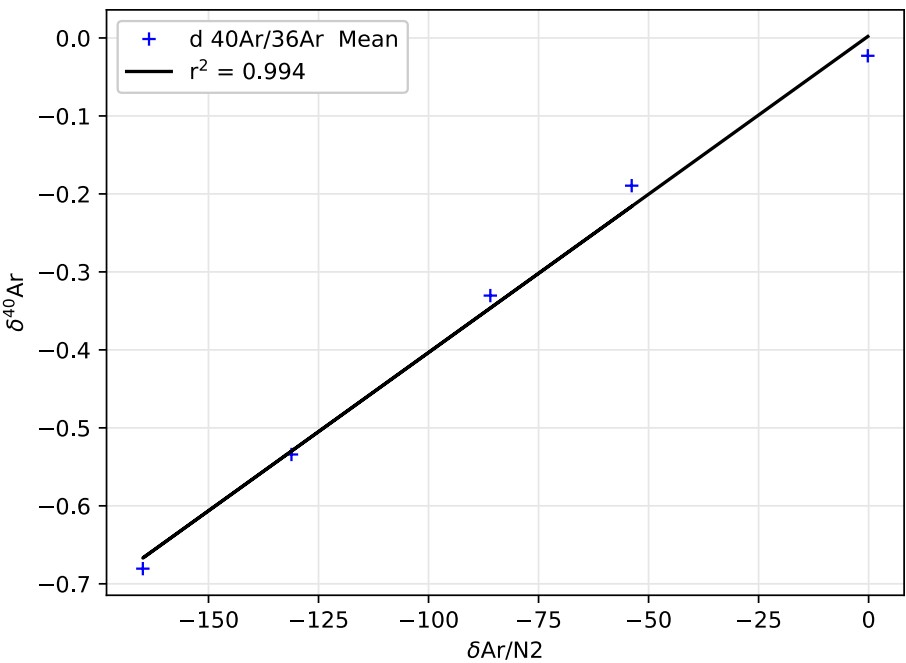


**Figure A9. Chemical Slope: variability of δ⁴⁰Ar versus δAr/N₂. Larger concentration of Ar in the gas mixture introduced into the spectrometer tend to increase the δ⁴⁰Ar values measured.**





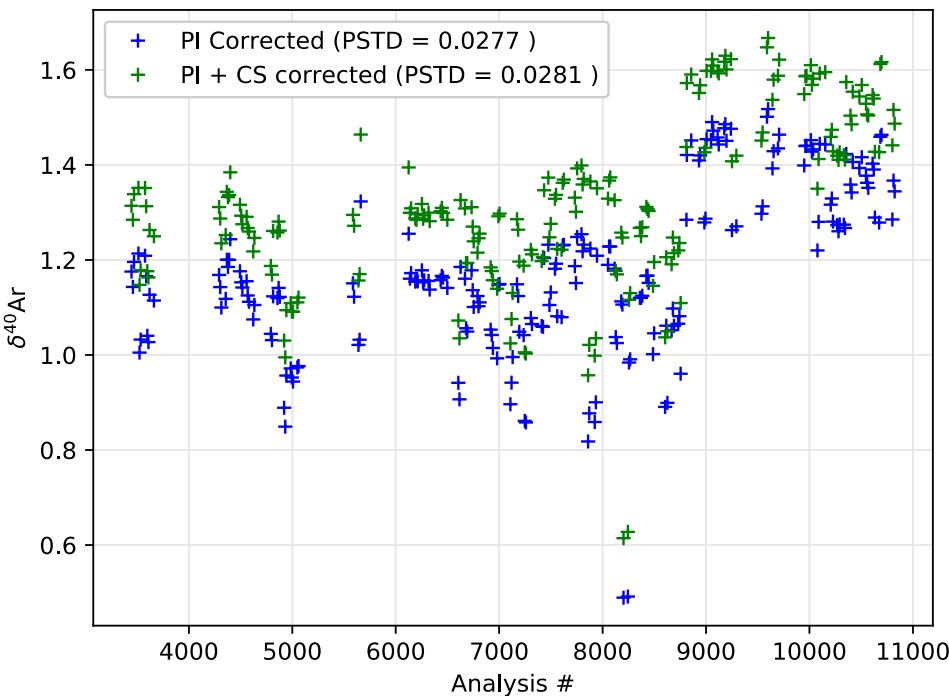

**Figure A10. Correction of the chemical slope effect on $\delta^{40}$Ar. Because the gas trapped in ice is generally argon-depleted compared to atmosphere, the chemical slope corrected values (green) are greater than non-corrected values (blue). Pooled Standard Deviation (PSTD) of each series are indicated in the captions.**

### Appendix A3.3 Drift correction

In addition to previously published correction, we identified that drifts around analysis 5000, or 7200 were caused by the spectrometer focus changing over time, causing a strong variability of $\delta^{40}$Ar and PIS. This drift could not be easily resolved because it only appeared after weeks of measurements. This issue was temporarily fixed with regular autofocus, but we needed a thorough recalibration of the focus parameters to make it more stable. Later, we fully recalibrated the spectrometer focus, causing a large change in raw $\delta$ values, both on samples and standards used for calibration (from the analysis number 8800). Fortunately, the focus changes were proportionally shifting the values of PIS and the $\delta^{40}$Ar, and did not affect $\delta^{15}$N values, which we attribute to nitrogen being the most abundant gas supplied to the spectrometer. To distinguish from potential pressure gradient-induced fractionation during sample preparation, which would affect both $\delta^{40}$Ar and $\delta^{15}$N proportionally to their mass, we removed the gravitational fractionation of $\delta^{40}$Ar using the $\delta^{15}$N. By doing so, the drift effect on $\delta^{40}$Ar could be quantified against the PIS$_{40\text{Ar}}$ as shown in Fig. A11. We define a drift slope as the non-gravitational fractionation of Ar versus the PIS$_{40\text{Ar}}$ in Eq. 5.

$$\text{Drift}_{^{40}\text{Ar}} = \frac{\Delta(\delta^{40}\text{Ar}_{\text{CS corrected}} - 4 \cdot \delta^{15}\text{N}_{\text{PI corrected}})}{\Delta \text{PIS}_{^{40}\text{Ar}}} \tag{5}$$

Using the drift slope, we then correct $\delta^{40}$Ar values:





$$\delta^{40}\mathrm{Ar}_{\text{drift corrected}} = \delta^{40}\mathrm{Ar}_{\text{CS corrected}} - \mathrm{Drift}\,_{^{40}\mathrm{Ar}} \times \mathrm{PIS}\,_{^{40}\mathrm{Ar}} \tag{6}$$

This improved the precision as attested by PSTD of $\delta^{40}$Ar in replicates that dropped from 0.028 ‰ to 0.013 ‰ (Fig. A12).
Note that if we apply the drift correction without the chemical slope correction, the PSTD only improves to 0.016 ‰ (not
shown), showing the importance of chemical slope correction prior to the drift correction.

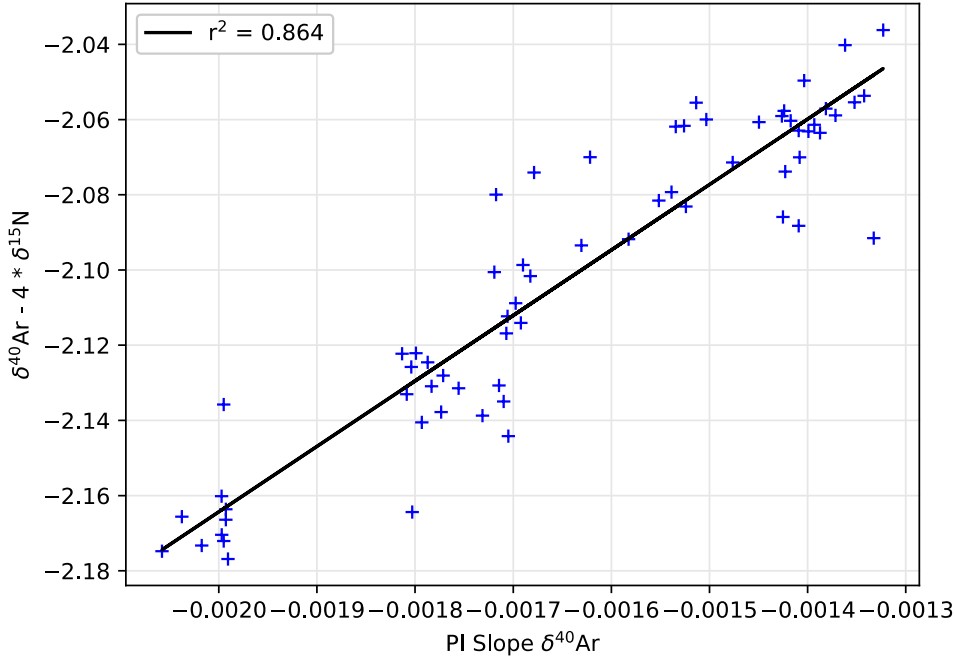


**Figure A11. Non-gravitational fractionation of argon as a function of the PIS for argon.**



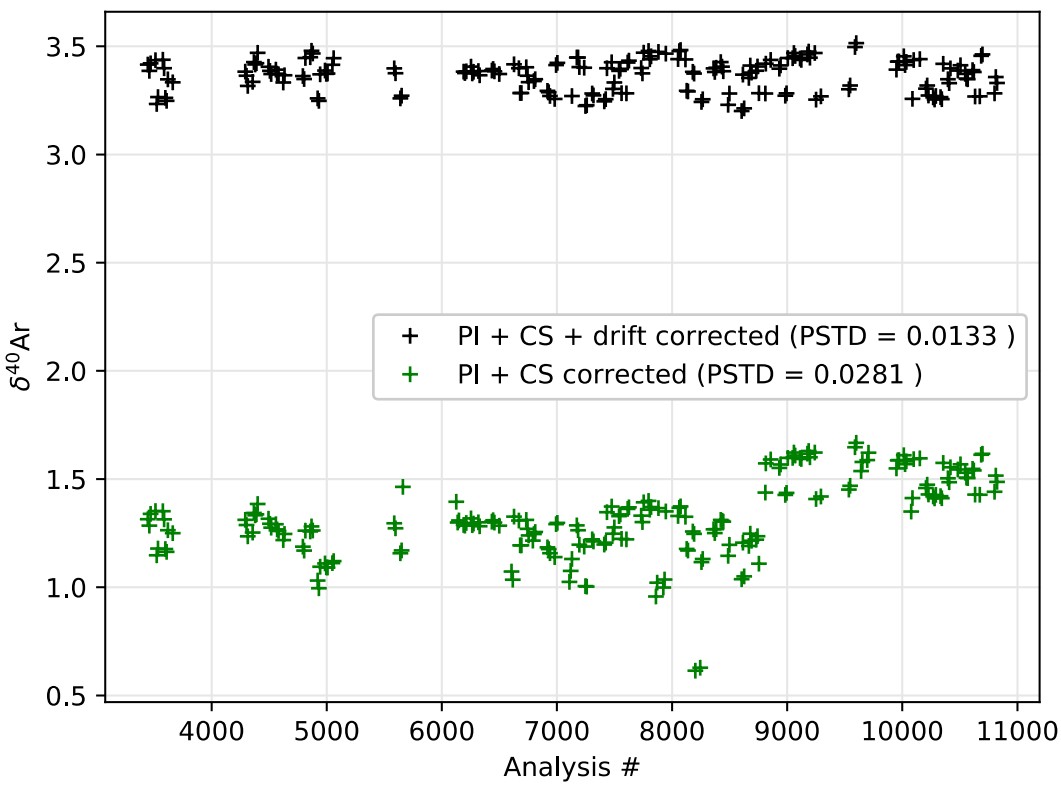

**Figure A12. Correction of the drift effect on $\delta^{40}$Ar. Pooled Standard Deviation (PSTD) of each series are indicated in the captions.**

**Appendix A3.4 Normalization to atmosphere**

All δ values are defined as the relative difference to a standard. For argon and nitrogen gases, the standard is the atmospheric air (IAEA, 1995). Two to three Air standards were prepared weekly from a bottle of air captured in Saclay in 2019. The same sample preparation and corrections are applied to the Air standard, as would be made for gas extracted from an ice sample. In total, 83 replicates of Air standards were measured, with a standard deviation after all corrections of 0.0052 ‰ for $\delta^{15}$N,

0.0188 ‰ for $\delta^{40}$Ar, and 0.0029 ‰ for $^{15}$N$_{excess}$. We believe that some of the error in the $\delta^{15}$N and $\delta^{40}$Ar is due to the sampling processing in the gas line, with gas being released into empty tubing, causing large pressure differences that are known to fractionate the isotopes. Even though we try to allow some time to equilibrate, a small fractionation probably remains. This affects the two sample replicates differently because they are processed separately, but both nitrogen and argon in the same sample are affected by the same pressure changes and resulting fractionation. Given that $^{15}$N$_{excess}$ is defined as the mass-

weighted difference between the two isotopic ratios, some of the fractionation induced by sample processing may cancel out, and therefore the difference in $^{15}$N$_{excess}$ between the two samples is smaller than $\delta^{15}$N or $\delta^{40}$Ar alone





We use weekly averages of two to three measurements of $\delta^{15}N$ and $\delta^{40}Ar$ in atmospheric air to normalize the ice measurements of the corresponding week (Fig. A13). By doing this correction weekly, we further correct any drift that can happen, as it would affect both our gas trapped in ice and atmospheric air standards in the same manner.

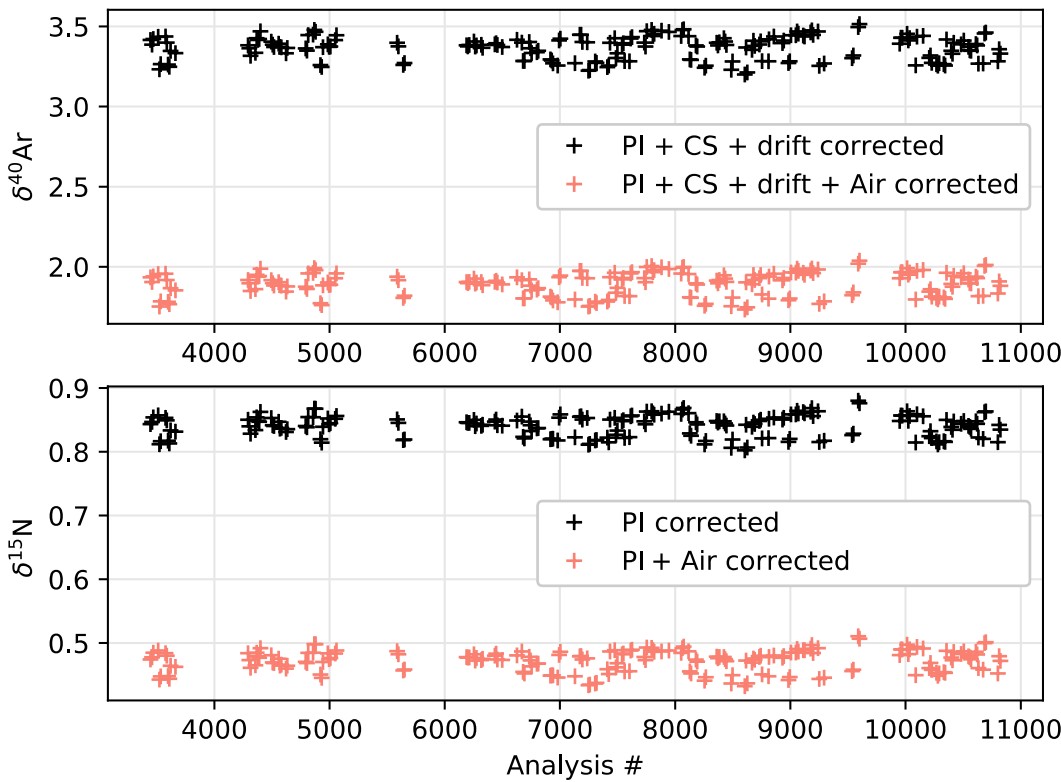


**Figure A13. Normalization to atmosphere. The values are subsequently given in ‰ versus atmosphere. Pooled Standard Deviation is unchanged with normalization because the same correction is applied for both duplicates.**

Normalization of the $\delta^{40}Ar$ and $\delta^{15}N$ of ice samples does not change the pooled standard deviation, because the two replicates are corrected by the same amount. However, it corrects the absolute value needed for quantification of gravitational and thermal

effects.

**Appendix A4 Advection-diffusion ice model tuning**

**Appendix A4.1 Parameters**

The model simulates the diffusion and advection of temperature in the ice. Given an initial state, the temperature will diffuse depending on boundary conditions such as surface temperature, advection due to accumulation of ice, compaction, bottom

temperature. We set the compaction to follow the ice density profile measured in the core, under the assumption that it did not change significantly in the past. Surface temperature is inferred from different temperature scenarios, plus a sinusoidal seasonal



cycle which amplitude of 13.7°C has been deduced from the temperature cycle at the ABN Automatic Weather Station. Bottom temperature is constrained by a constant geothermal heat flux of 65 mW m$^{-2}$ for Aurora Basin (Maule, 2005; Martos et al., 2017). Temperature diffusion simulations start at 2000 BCE and last for 4014 years, allowing for flexibility in the first ~2000
years where we have no gas constraints. For the past 2700 years (700 BCE onwards), we use the accumulation obtained with annual layer counting with a 20-year running average (as shown in Supp. Fig. 3). Previous this date, we use the average accumulation of 94 kg m$^2$ a$^{-1}$ (Akers et al., 2022). The model is also highly dependent on the initial temperature profile in the ice.

The ice thickness is suspected to have varied over time both spatially and temporally. Upstream ABN coring location, the ice
about is 500 m thicker than the current 3500 m, mainly because bedrock is deeper in the central part of Aurora Basin. We assume that temporal variations of ice thickness in Aurora Basin during the past 4000 years are much smaller than this. For simplification, we use an averaged thickness of 3700 m over the past 4000 years. Changes of ice thickness of 500 m caused by a surface change of accumulation may affect the temperature gradient in the upper ice column (Morgan et al., 2022), but thickness changes associated with glacial flow should not have a significant impact at the scale of 2000 years, because most
of the ice deformation occurs at depth (Wang and Warner, 1999; Doyle et al., 2018), and equilibration time between the bedrock and the surface is of the order of magnitude of 10,000 years. The accumulation history used in the model is inferred from the ice core accumulation with a 20-year smoothing.

Finally, the diffusion model outputs gradients of temperature between the surface and a lock-in depth, which is inferred from the lock-in depth history determined in Sect. 3.2 of main article. Uncertainties of ± 4 m in the lock-in depth result in ± 0.03°C
in the surface to lock-in depth temperature gradient, ten times smaller than the uncertainties on the ΔT estimated from the gases.

**Appendix A4.2 Initialization with consideration of ice flow**

We account for long-term influence of ABN site displacement on the temperature scenarios using a transient state initialization. Most deformations associated with ice flow occur in the bottom layers of ice in contact with the rocks, where the temperature
is near melting point (Wang and Warner, 1999; Doyle et al., 2018). The uppermost column of ice is advected rather uniformly, so we can consider simply consider surface temperature changes from a Lagrangian point of view, where we follow the column of ice under ABN as it flows. Therefore, we consider two components of temperature changes: spatial temperature changes due to the displacement of ice, and climatic temperature changes. For simplification, and because our simulation only spans the last 4000 years, we consider the surface topography as constant, and assume that climatic temperature changes vary
homogeneously spatially in the range of 40 km. This allows us to simply run the diffusion-advection model with a single dimension (depth).

We chose to initialize the model with a transient-state profile rather than a stationary temperature profile to simulate the presence of cold ice advected under ABN coring site, and better match the −2.5°C temperature gradient observed in the





borehole measurement. The initialization is performed with a transient state temperature simulation, in which we equilibrated

the ice column with a surface temperature of −61°C for 20,000 years and then simulate the deglaciation and the Holocene with

the temperature history from Dome C (Jouzel et al., 2007) and ice-flow related temperature changes from modern surface

temperature (Agosta et al., 2019) and ice flow velocities (Mouginot et al., 2019). The temperature history used during

initialization is shown in Fig. A14. The initialization run is unique and starts from 30,000 BCE to 2,000 BCE. The past

accumulation during initialization is set at 40 % of the average ABN accumulation, increasing to 80 % during deglaciation and

ramping up to 100 % at 2,000 BCE. The ratio of accumulation between the last glacial maximum and the late Holocene is set

to be roughly the same as Dome C (Wolff et al., 2010) and WAIS Divide (Buizert et al., 2015). Arguably, the ice-flow was

slower than modern day in the glacial period, which would result in smaller temperature changes if modelled correctly, but

such precision is not needed in the initialization of the model. This transient initialization produces a temperature profile with

a strong gradient, which can match the observations within 0.5°C if the simulation is continued until 2014 CE (Fig. A15), and

is thus a good starting point for the temperature simulations.

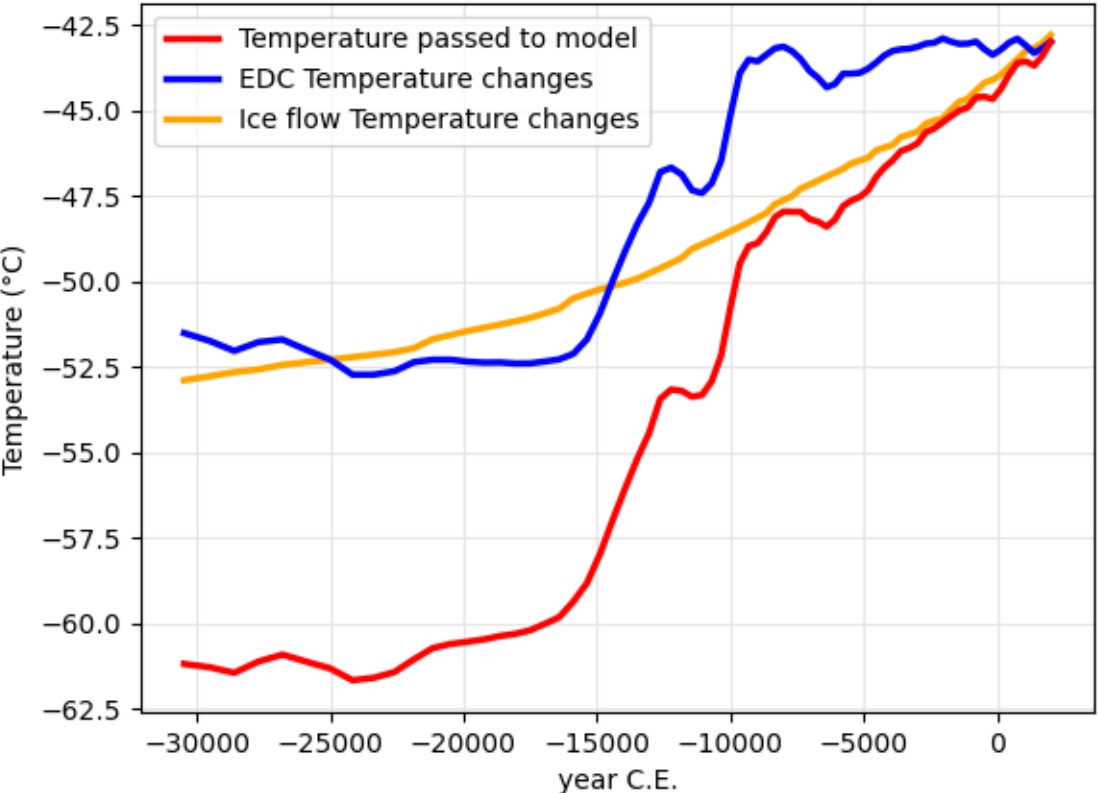

**Figure A14. Base surface temperature passed to the model in the transient state initialization, which runs from 30,000 BCE to 2000 BCE, and the unperturbed simulation from 2000 BCE to 2014 CE. Red line shows the total temperature changes, blue line shows**



**EDC temperature changes (Jouzel et al., 2007; shifted to match ABN modern temperature) and orange line shows the ice-flow related**
**temperature changes from a simple interpolation of modern-day Antarctic surface temperature along the estimated back-projected**
**position of ABN using modern flow speeds (Mouginot et al., 2019).**

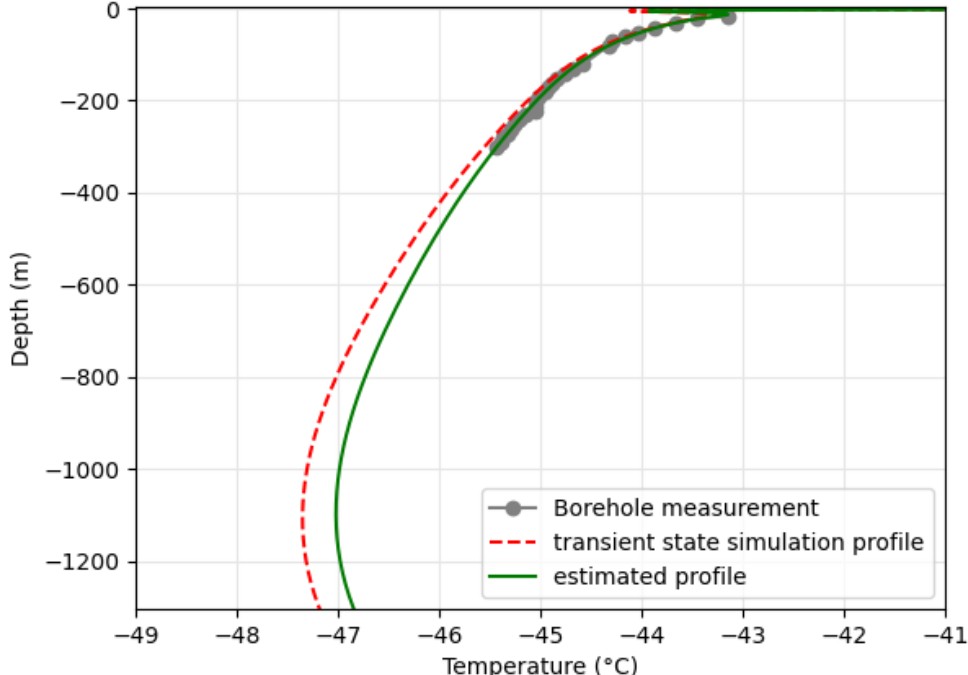

**Figure A15. Temperature profile obtained after the transient state initialization (red dotted line), as compared to the borehole temperature profile (grey) and the estimated temperature profile after inversion (green).**

**Appendix A4.3 Temperature perturbation simulations**

Starting from the transient state profile of temperature at 2000 BCE, we start simulations following the same base surface temperature forcing (later part of Fig. A14), and add a "triangle" shaped perturbation in each simulation, with an additional temperature increase ramping up to +1°C. The added perturbation increases over the course of 10 years then decreases in another 10 years to return to the base surface temperature forcing for the remainder of the simulation. The next simulation uses
a perturbation that is shifted in the time by 10 years. A few longer lasting perturbations are used in the 2000 BCE – 0 CE period, where the constraints are low because of the absence of gas data, and the climate information is not used in the results of this study. We perform a total of 220 simulations with perturbations at different times in the run.



**Appendix A5 Parameters of the temperature inversion**

**Appendix A5.1 Shift of the $^{15}N_{excess}$ mean value**

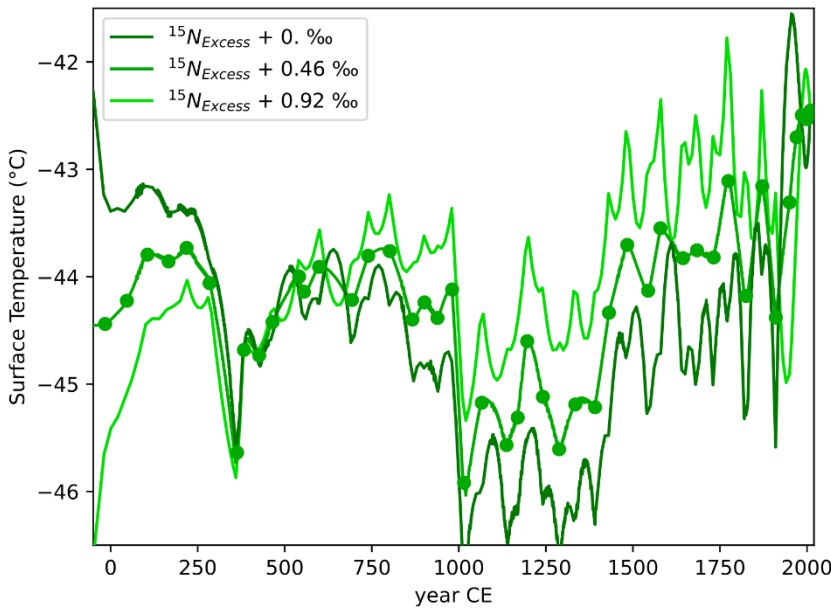

**Figure A16. History of temperature obtained with inversion of borehole and temperature gradients obtained with $^{15}N_{excess}$, using different correction values for $^{15}N_{excess}$. Not correcting or correcting too strongly the $^{15}N_{excess}$ causes the temperature reconstruction to squeeze fast saw-tooth shaped variations between ages where we have gases datapoints, represented by the dots.**

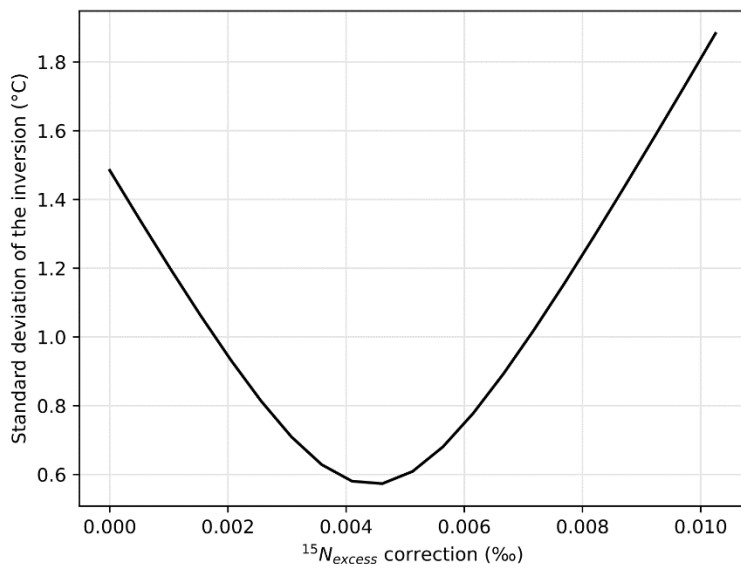


**Figure A17. Standard deviation of the history of temperature depending on the value used for $^{15}N_{excess}$ correction. A value of 0.0046 ‰ is best to minimize the intrinsic variability of the reconstruction and match both datasets.**





**Appendix A5.2 Uncertainty parameters**

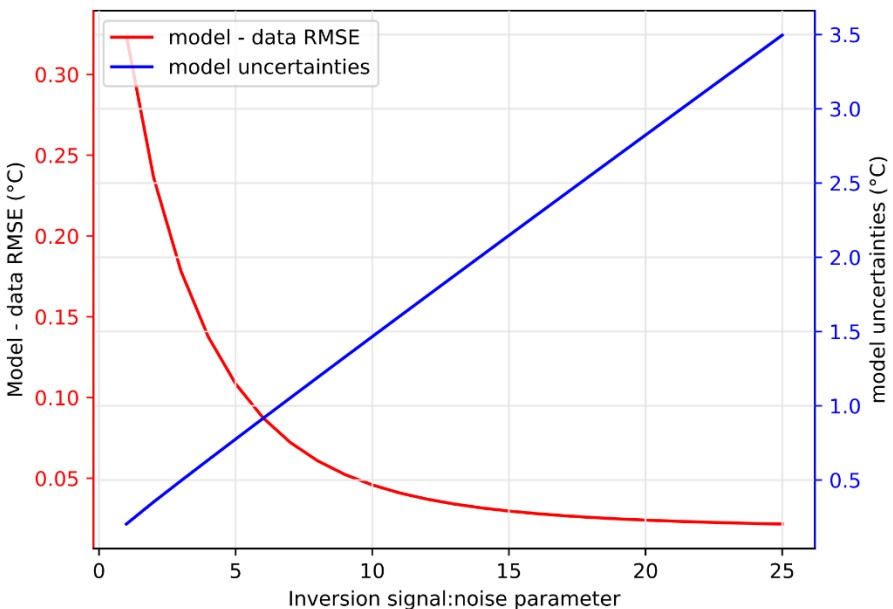

**Figure A18. Root Mean Square Error (RMSE, red) between temperature reconstruction and temperature data passed to the inversion, and reconstruction uncertainty (blue) as a function of signal to noise parameter. A signal to noise ratio too low results in a too smooth inversion with high model-data mismatch, whereas a too high parameter increases the resulting inversion uncertainty to allow all for more flexibility**

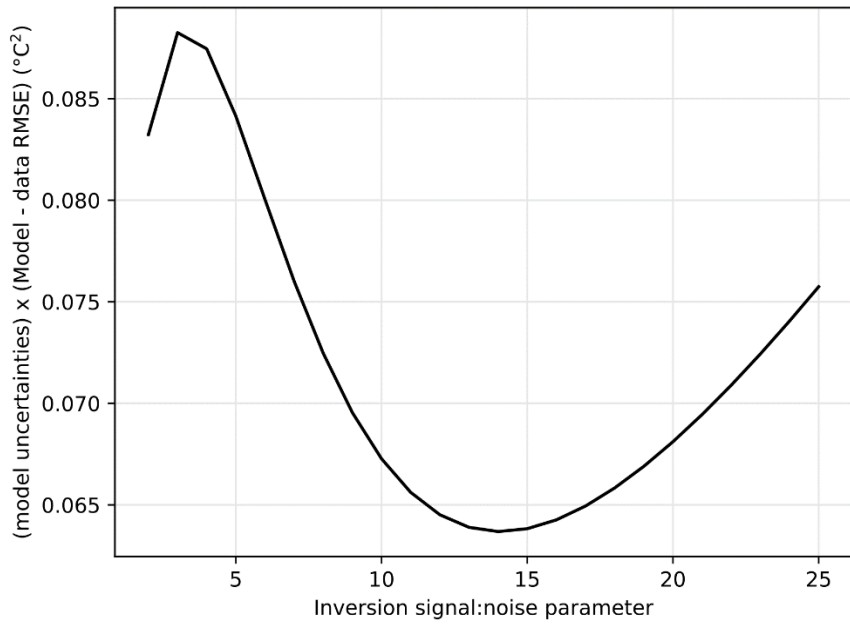

**Figure A19. Cost-function of the reconstruction as a function of signal to noise parameter used in the inversion. A value of 14 is best to ensure that the inversion matches the constraints and minimize its uncertainty.**




**Appendix A5.3 Age significance of data points in the reconstruction**

The diffusion of temperature in the firn depends both on snow and ice properties but also the accumulation rate at the ice core site. At West Antarctic Ice Sheet Divide (WAIS Divide), the temperature measured at depths corresponding to the last
millennium was estimated to integrate temperature with a smoothing window of several hundred years (Orsi et al., 2012); for ABN where the ice accumulation is half as high as WAIS Divide, we expect even greater smoothing windows. The temporal range of temperature changes recorded by $^{15}N_{excess}$ at ABN is limited at the minimum by the diffusion time of gases between the surface and the lock-in zone, and at maximum by the smoothing out of the temperature gradient by the temperature diffusion in the ice column. At NEEM, gases take on average 10 years to diffuse down to the lock-in depth of 63 m (Witrant et al.,
2012); we expect that at ABN, where the lock-in depth is about 30 m deeper, the gases will take slightly longer to equilibrate in the diffusive column, so a temperature gradient maintained approximately 20 years should be effectively recorded in the $^{15}N_{excess}$.

In addition, we further reduced the time-resolution of gas-based reconstruction by averaging the isotopic values in 5 m windows, with an expected age range of about 45 years. Lastly, we applied a smoothing in the inversion to limit the variability
of neighbouring points by forcing a covariance in the reconstruction. The covariance parameter was set to be exponentially decreasing with age difference, with a 0.5 covariance reached at 70 years difference. Therefore, the reconstruction includes multiple gas data points for each inversion point, strengthening the confidence in the reconstructed temperature. In our figures, we chose to represent the inversion at the time-resolution of points when there is a gas-based temperature constraint.

**Data availability**

Water isotope ($\delta^{18}O$ and $\delta D$), gas isotope ($\delta^{15}N$ and $\delta^{40}Ar$), borehole temperature data, as well as temperature reconstructions will be accessible in a repository on the Australian Antarctic Division data centre website (https://data.gov.au/ – full link will be available upon acceptation of the present manuscript).

**Author contribution**

MC directed the ABN ice core drilling project. AO and AS designed the study. Raw water isotope data was measured and curated by AM. AO designed the gas isotope analysis, AS performed the gas isotope analysis. TP measured the borehole temperature. AO secured financial resources for gas laboratory analyses and, together with AL, supervised stable isotope interpretations. JMC measured water stable isotopes in the short core used for calibration. JMC, XF and JC measured the methane used for gas age model. ELM provided upstream GPS elevation profiles and radar reflector surfaces. CP performed
the annual layer counting, MC provided volcanic tie ages for ice age model. AO adapted the ice diffusion model and designed the inversion algorithm. AS performed the diffusion model simulations. AS and AO interpreted the climate implications of the



results. AS wrote the original draft and created the figures. AP, AJ, MC, AM, AL, JMC, ELM, XF, and JC contributed to the review of the manuscript.

## Competing interests

The authors declare that they have no conflict of interest.

## Acknowledgments

The authors would like to thank the Aurora Basin North field team, in particular Simeon Sheldon for the help with the borehole temperature measurement, and Christopher Plummer for help with annual layer counting. The Australian Antarctic Division provided funding and logistical support (AAS project 4075). Additional logistic support was provided by the French Polar
Institute IPEV through the CHICTABA project n°1115. Measurement of gases isotopes at LSCE were supported by CNRS-INSU LEFE project ABN2k.

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
