# Peer review of "A 2000-year temperature reconstruction on the East Antarctic plateau, from argon-nitrogen and water stable isotopes in the Aurora Basin North ice core."

_Climate of the Past, 2022_

## Author Comment (AC1)

**Note: Comments from the referee are underlined, and our response follows.**

We appreciate the suggestions given by Referee#1, that helped clarify the conclusion of the article. We understand that some statements in the discussion on teleconnection lacked argumentation and should therefore be removed. Referee#1 stated that "A model simulation would be required to test this hypothesis"; this would be a possible alternative to discuss the influence of climate modes and teleconnections, but is too long to fit in this article.

Nonetheless, we would like to discuss the temperature at the millennial scale in comparison to the reconstructed SAM (Southern Annular Mode, or the zonally symmetric variability), but with more restraint.

**"Major comments"**

**"Section 4.3 about teleconnections is too speculative. […] The SAM and the temperature reconstruction in Figure 11 do not resemble one another. Even the long-term trends oppose."**

SAM reconstructions have been made for the last 1000 years, mostly from temperature-dependent proxies of sites under the influence of SAM. Although the SAM description can only be made for instrumental periods when geopotential variability is constrained by pressure measurements, its influence on temperature has been the subject of previous studies, on which we rely to analyze our temperature record.

For the purpose of discussion with referees, we studied the correlation between SAM reconstruction and temperature at ABN (Figs. R1 & R2). We observe a significant negative correlation between SAM and $^{15}N_{excess}$ temperature on the period 1000-1900 CE (Fig. R1a). However, when including the points later than 1900 (Fig. R1b), the correlation disappears as both Temperature and SAM phase increase. It is difficult to argue that the 20[th] century is the exception because calibration of SAM index for the reconstruction was made mostly with data covering the 20[th] century. The correlations are not significant between the SAM and $\delta^{18}O$ temperature (Fig. R2). We thus substantially toned down the teleconnections interpretation, and removed discussion about PSA2.

Detailed changes: In lieu of Section 4.3, we would consider adding the following paragraph to Section 4.2, whose title will be changed to "Climate implications". Discussion on WAIS Divide and Taylor Dome ice cores to discuss PSA2 variability will be removed, so we will remove the graphs **c** and **d** from Fig. 11 as well. Trends will no longer appear on Fig. 11.

"In the Southern High Latitudes, the Southern Annular Mode (SAM) describes the main mode of geopotential variability (Limpasuvan and Hartmann, 1999), led by meridional pressure differences (Gong and Wang, 1999). This results in zonally symmetric variability with a visible effect on Antarctic surface temperatures (Broeke and Lipzig, 2003). On the East Antarctic Plateau, SAM phase and surface temperature are anti-correlated because a stronger meridional pressure gradient is associated with reduced poleward heat transport (Marshall and Thompson 2016), and the SAM signature is found in the temperature at the ABN site, but SAM does not affect $\delta^{18}O$ significantly (Servettaz 2020). On the timescale of a thousand years, the annual SAM has been reconstructed from paleoclimate proxies sensitive to SAM-related temperature anomalies (Datwyller 2018; Fig 11b). The $^{15}N_{excess}$ temperature cold interval during 1000-1400 CE cooccurs with a positive phase of the SAM, then the shift to strongly negative SAM accompanies the warming at ABN between 1400 and 1500 CE. While this temperature

pattern matches the SAM variability, the temperature evolution over the latter half of the last millennium is not explained by SAM changes, as both $^{15}N_{excess}$ temperature and SAM follow an increasing trend. SAM may play a role, but is not clearly the only source of surface temperature variability."

[Figure]

Figure 11 (simplified): (a) δ18O temperature and 15Nexcess temperature reconstructions (this study). Error shades are the same as in Fig. 9. (b) Southern Annual Mode (SAM) annual reconstruction (Dätwyler et al., 2018). Thin lines show the annual reconstruction, thick lines are the 30-year average for both δ18O temperature and SAM; $^{15}N_{excess}$ temperature has a resolution of about 45 years. Yellow shading highlights the 1000–1400 CE period during which the 15Nexcess temperature is significantly colder, in phase with a positive SAM index.

[Figure]

Figure R1: Scatter-plots and linear regressions for SAM annual index (Dätwyler 2018) and $^{15}N_{excess}$ temperature for the 1000-1900 CE period (**a**) and the full period (**b**). SAM index was averaged on the $^{15}N_{excess}$ temperature resolution.

[Figure]

Figure R2: Scatter-plots and linear regressions for SAM annual index (Dätwyler 2018) and $\delta^{18}O$ temperature for the 1000-1900 CE period (**a**) and the full period (**b**). SAM index was averaged on the $\delta^{18}O$ temperature resolution (30 years).

**"The 1991 end year seems arbitrary, why not go back to 1979, or calibrate with 2-year averages if the age scale isn't reliable at deeper depths?**

We will further justify: "Annual layers could be identified down to the Pinatubo eruption (1991). Below this depth, uncertainties on the dating do not allow for clear annual averages, and multi-year average could lessen the range of variability, therefore we calibrate a δ18O – temperature slope for ABN using

linear regression on the 1991 to 2013 period." And delete "where we are confident on the dating and have a decent number of years"

**Fig 6b the r^2=0.316 value seems low. How does this compare to other studies?"**

I am unsure whether annually averaged snow isotope vs temperature correlations are frequently reported in snow studies. For correlation at higher temporal resolution, Steen-Larsen et al. (2014) reported pearson r values of 0.17 (r^2 = 0.03) in 2011 and 0.32 (r^2 = 0.10) in 2012 for NEEM, Greenland. Similarly, Casado et al. (2018) reported surface-snow correlations of r^2 = 0.29 for Dome C in 2011. Note that the correlations of temperature with snow isotopes are much weaker than with precipitation isotopes, because of signal modulation by precipitation intermuttency and post-deposition modification of signal. In any case, we show the 95% confidence intervals for the slope value as dashed lines for Fig. 6 and shading for Figs 9 & 11, to represent the uncertainty on the slope.

**"Is it correct to use the 2mT from MAR? Consider using a simple Rayleigh-type model (e.g. SWIM) instead to reconstruct surface air temperature at the core site (Markle and Steig 2022; Jones et al. 2023)."**

Owing to its detailed near-surface vertical resolution, the MAR model excels at reproducing surface temperature. It is more accurate than taking climate re-analysis. We discussed the performance of MAR at GC41 Automatic Weather Station near ABN site in the supplementary of the Servettaz et al. (2020) article. Since there was no Automatic Weather Station at the exact ABN location prior to the drilling campaign, MAR is the best temperature data that we can use for experimental definition of a slope.

We did consider defining the slope using different models including Rayleigh-type model, however we think this model cannot account for deposition dynamics of snowfall (changes in seasonality and infrequent precipitations do not represent the year-round average) and post-deposition effects which could substantially change the snow isotopes and thus the slope value. This is based on a comparison of Mixed Cloud Isotopic Model (Rayleigh-type model), precipitation-based, and surface snow-based slopes presented in Casado et al. (2018), who found slope values to differ significantly. In our understanding, the benefit of using SWIM model is that temperature-isotope relationship can be non-linear and more quantitative. However, non-linearities in the SWIM model (Markle and Steig 2022) are concentrated near the positive condensation temperatures, which are not expected at a site as cold as ABN (temperature above inversion, where condensation occurs, was estimated at 29.7°C; Servettaz et al., 2020). We thus concluded that using the SWIM model for a time-range of 2000 years, with little change in source temperature, and for condensation temperatures contained between -40 to -20°C would not significantly differ from using a single slope value; please tell me if this is incorrect.

Originally, we were planning to use ECHAM model to defne a slope for surface temperature similarly to how Stenni et al. (2017) defined slopes for various Antarctic ice cores (we were studying slopes from ECHAM5-wiso in Servettaz et al. 2020). The advantage of using ECHAM model is that we can estimate surface-snow as accumulated from irregular precipitation events and account for the precipitation dynamics to some extent, and it can account for evaporation source changes as well. However, this still does not account for post-deposition effects on $\delta^{18}$O. We were trying to improve from this model-defined slope by using high resolution in-situ $\delta^{18}$O measurements.

Regarding the slope value, which is quite high in comparison with models, we have recently put some effort to understand the origin of slope "steepening" when averaging annually. Our hypothesis is that the over-representation of warm events in a year of snow can make the annual $\delta^{18}$O appear much higher, when the yearly temperature is slightly warmer than average. This effect substantially increases the $\delta^{18}$O-T slope value if we define it from yearly averages. This will be discussed in a future article, currently under writing. We still think it is best to use the slope based on annual averages, because the measurements of $\delta^{18}$O in the ice core do represent yearly or multi-year averages. This is the easy workaround to avoid the warm bias of winter events, because through the "steep" slope, even large $\delta^{18}$O excursions will have a limited reconstructed temperature effect. Note that the steep slope is not the reason why we do not see the cold period for 1000-1400 CE with the $\delta^{18}$O, but rather because the $\delta^{18}$O values are not lower during this period.

**"Is there a winter bias in precipitation (Servettaz et al. 2020)? So, the d18O temperature reconstruction would be a winter temperature record. Perhaps it can explain the record being less variable. The winter WDC isotopes record was less variable compared to the summer and annual means (Jones et al. 2023)."**

There is indeed more precipitation in winter on average, but the winter temperature is also more variable, depending on the occurrence of warm events (Servettaz 2020). The conclusion we drew from this stable $\delta^{18}$O record is that: processes implied in precipitation reaching ABN did not change in temperature, but may have become more infrequent in the winter season. This widens the gap between temperature recorded in the $\delta^{18}$O representative of a few precipitation events, and the $^{15}$N$_{excess}$ which averages many years of surface temperature. This hypothesis is presented in Sect 4.1 Lines 491-505, but it will need to be further studied in an article dedicated to the dependence of average $\delta^{18}$O on frequency and timing of precipitations in a year.

**"In Servettaz et al 2020, you argue that SAM- is associated with d18O peaks. However, no significant correlation with SAM is provided in either paper. The logic seems flawed since the trend in SAM is towards a more positive phase and the isotopes appear to display no trend or positive trend over the recent past, which doesn't fit with the negative SAM argument in your previous paper. The trend analysis period seems arbitrary (Fig. 11). If you start the trend analysis around 1500 to present instead then you would get a positive SAM trend but there is no clear change in the isotopes over this period. From what you have presented there is no clear evidence that the isotopes are driven by SAM. This is too speculative, so it needs to be removed if no additional supporting analysis is provided."**

As answered above and represented in Figs. R1-R2. Discussions were based on 1000-1900 CE trends and correlations on the basis that "natural variability" was occurring in the pre-industrial era, before the post-1900 anthropogenic warming. Since no significant correlation exists over the full period, we removed text attributing temperature or isotope variability at ABN to SAM.

**Could you use spectral analysis to check if there is a SAM and ENSO signal in d18O?**

We conducted spectral analysis on SAM and $\delta^{18}$O from ABN (Fig. R3), using a multi-taper method package for Python, described by (Prieto, 2022). We represent the Power Spectral Density (PSD) as a function of revolution period (Fig. R3c and d), as well as F-test statistic for periodic components (Fig. R3e and f).

For ENSO, the $\delta^{18}O$ resolution may be too low with the 20-cm sampling presented here (average accumulation is ~10 cm ice/year, so the power spectral density starts at 4 years). In any case, no periodicity strongly stands out on the PSD figures, and the F-test statistic returns no matching periodicities between the SAM and $\delta^{18}O$. F-test statistics indicate that some power around the 6-year periodicity is important, which could be related to ENSO, but this is not very clear in the PSD and would require more investigation to confirm.

[Figure]

Figure R3: spectral analysis of $\delta$18O (**a**, **c**, **e**) and SAM annual index (**b**, **d**, **f**, Dätwyler et al. 2018), including time-series (**a** and **b**), Power Spectral Densities (**c** and **d**), and F-test statistic for periodic components (**e** and **f**).

**"Why is the summer SAM index displayed (Fig. 11)?"**

The annual SAM was indeed used, as indicated in the text Line 627. The axis is mislabeled, but annual data was correctly displayed (the two datasets are provided in Dätwyler 2018, but we finally selected the annual index to compare with $\delta^{18}O$ and $^{15}N_{excess}$ data which are not season-restricted). We will rectify the axis label.

**"Discussion section 4.1. Some of the discussion here is too conversational without backing up with supporting test results. Aim to be concise in the revised version."**

We shortened some paragraphs by removing unnecessary information, and further justified the paragraph discussing the influence of topographic slope and wind speed on temperature (see Fig. R7 in the response to Referee#2 for details on the slope-temperature discussion):

[revised manuscript text omitted]

Similarly in the conclusion, removed:

L679: "The surface warming at ABN after 1400 CE contrasts with West Antarctic $\delta^{18}O$ records and indicates the influence of zonally asymmetric Pacific-South American atmospheric modes." Replaced with "The warming trend from the second half of the last millennium while SAM phase is increasingly positive implies a temperature control through other mechanisms as well."

L684: ", as shown by its remarkable consistency with SAM variability"

**"L158. Remove the first 'and'. Check for this type of typo in the whole document."**

Several sentences were rephrased:

Line 140 "Aliquots of water were sampled by a Picarro liquid auto-sampler, injected into a Picarro high precision vaporization module (A0211), and held at temperature of 110°C, then vapour is sent to the Picarro L2130-i isotopic water analyser"

Line 158 "the ice was melted in a pre-emptively evacuated bottle, the gases were released in a processing line with cold traps to remove water vapour and carbon dioxide, and a heated copper mesh (500°C) to remove molecular oxygen"

Line 257 "Firn characteristics may vary through time, affecting the height of the diffusive zone and thus the lock-in depth, hence the gas age model is further refined with the methane record measured in the ABN1314 core"

Line 280 removed "Fig. 4 shows The gas and ice age models and the difference of age between gases and ice at a given depth." Added instead references to Fig. 4 Line 269: "the gas age model of ABN1314 core only covers the last 2050 years (Fig. 4)" and Line 281: "The gas-ice age difference at a given depth (Fig. 4b) is comprised between 600 and 700 years (…)"

Line 457 changed second and to "from": "To account for this effect, we consider the ice in the diffusion-advection model in a Lagrangian perspective, and dissociate temperature changes caused by site displacement from climatic temperature changes"

Line 481 changed second "and" to "as well as": "with cold periods from 300 to 450 CE and from 1000 to 1400 CE, as well as a recent warming of about 1°C."

Line 486 (Fig. 9 caption) rewritten as "Comparison of δ18O temperature and $^{15}N_{excess}$ temperature reconstructions (**a**) with upstream elevation (**b**), upstream slopes (**c**), and $d_{ln}$ (**d**) in the ABN1314 ice core."

Line 540 changed to semi-column to avoid repetition of "and": "in absence of mixing of air in the surface layer, the winter temperature inversion cools the snow surface; this densifies the near-surface firn air which could sink and advect the air column downward more efficiently than during summer"

Line 596 rephrased to "This surface warming at ABN is unlikely to be caused by a topographic change as the flattening slope near the drilling site (Fig. 9) would on the contrary favour the slowing of katabatic winds and surface cooling by strengthening of the near-surface temperature inversion"

Line 850 rephrased to "we equilibrated the ice column with a surface temperature of −61°C for 20,000 years, then simulate the deglaciation and the Holocene with the temperature history from Dome C (Jouzel et al., 2007), added to ice-flow related temperature changes calculated from modern surface temperature (Agosta et al., 2019) and ice flow velocities (Mouginot et al., 2019)."

**"L 194. Change the word 'thinly'."**

Changed "thinly closed" to "enclosed by a thin ice wall".

**"L 203. Instead of calling it 'resampled' call it a 5 m moving average. As with resampled taking every 5th m value comes to mind. Change throughout."**

L202 "by resampling using a 5 m window" changed to "by averaging on 5 m windows".

L203 "resampled" changed to "averaged"

**"L230. Previously, you wrote that the water isotopes from the ABN1314 core were measured discretely on a Picarro. Here you state that they were measured on a CFA system. I guess they were measured on two setups at two labs but be clear in the manuscript."**

Added for precision: "Although water isotopes are also measured on CFA system, the CFA data was only used to build the age model; in this article we discuss the isotope data from discrete sampling measured at the Australian Antarctic Division (Sect. 2.2.1)."

**"L245. Perhaps use the word peak instead of 'extremum'."**

"Extremum" replaced by "peak"

**"Number the appendices in the order of appearance in the text?"**

Appendix A3 is called first and thus moved to top -> Appendix A1.

Appendix A1 & A2 become A2 & A3 respectively, and are called through Fig. numbers rather than Appendix Section because they are not accompanied by text.

Accordingly, Figs. A7 – A13 number change to Figs. A1 – A7, and Figs A1 – A6 change to A8 – A13.

**"How were the short-core isotopes measured? Provide more information about the short core, dating, and which range of years it covers. As the start year for the calibration 1991 isn't the same as the start of the overlap with the satellite era, you cannot call the range in Fig. 6a "their overlapping period" as the full period is not used."**

We precised "Water stable isotopes in the shallow core were measured at high resolution with Continuous Flow Analysis (CFA) at the Desert Research Institute (Servettaz et al. 2020)."

For the start year of 1991, see the discussion above. Changed "on their overlapping period of 23 full years from 1991 to 2013" to "on the 1991–2013 period"

**"Define the isotopes and which international standards were used."**

Water isotopes standard is given L143 and gas isotope standard is given L168. Added a reference to IAEA in L140 after "Water stable isotopes (noted $\delta^{18}O$ and $\delta D$, as in IAEA 1995)"

**"L603. I wouldn't call it 'many', as there aren't that many ice core sites on the plateau. 'The more abundant'?"**

I think we can just remove many and keep "The 15Nexcess and borehole temperature reconstruction provides new insight on the climate of East Antarctica that may complement the $\delta^{18}O$ records in this region."

**"L605. D18O is perhaps a proxy for winter temperature while the other represents annual temperature. Therefore, you cannot make a judgment on which proxy is best."**

Added annual in the following sentence to be more correct: "Together they consolidate the evidence that annual surface temperature changed with a greater amplitude than what δ18O suggests".

**"L608. Define SAM and the meaning of the SAM acronym at first mention in the text (L599)."**

This section has been reworked and reduced to a single paragraph, as discussed above. SAM is defined at its first occurrence.

**"L608. Marshall and Thompson, 2016 were not the first with discovering SAM's significance on the Antarctic climate. Provide more references."**

Although Marshall and Thompson are not the first to describe the SAM, their study is recent and impacts of SAM on temperature are clearly assessed, which is why we initially relied primarily on this citation.

We added references to historical papers on Antarctic Oscillation / SAM (Limpasuvan and Hartmann, 1999; Gong and Wang, 1999; Broeke and Lipzig, 2003).

**"Add a paragraph that describes the model and reanalysis data that was used. State which organization provides the MAR data and reference it. And that it is a high-res model for the plateau driven by ERA-interim as you did in (Servettaz et al. 2020)."**

We propose to detail L323-L326: "We chose to determine the ABN δ18O – temperature slope using the $\delta^{18}O$ record from the 12 m shallow core described in Servettaz et al. (2020) and temperatures from the regional atmospheric climate model MAR (available at https://mar.cnrs.fr/, we use a simulation described in Agosta et al., 2019) nudged to ERA-Interim climate reanalysis (Dee et al., 2011). The MAR model was developed with implementation of specific physical parameterizations for polar regions, with a turbulent scheme adapted for stable conditions of the Antarctic Plateau. It has a high vertical resolution with five levels within the first 10 m, which enables a good representation of temperature inversion. Consequently, MAR was shown to model the surface temperature more accurately than any other available dataset when compared with automatic weather station observations near ABN, with a bias lower than 1°C (Servettaz et al., 2020)."

**"Figures"**

**"Fig. 6. Remove the DRI acronym or use it throughout and define it at the first mention (L231)."**

Changed to "Shallow Core"

**"My personal preference would be that you call the core "shallow core" instead of short core. Like you did in your previous paper (Servettaz et al. 2020)."**

We changed the occurrences of "short core" to "shallow core" in Fig. 6 caption and label (L333), and author contribution (L923).

**"Fig. 7. Only orange shading is shown in the plot."**

It was the result of compilation error, the blue shading for negative values will be correctly displayed. Sorry for not picking up this. Figures will be carefully checked in the next version.

**"Fig. 8. The line is gray, not black."**

Changed the caption to "grey dashed line"

**"Fig. 11. Why is the SAM summer index displayed? Display annual index values instead."**

The annual SAM was indeed used, as indicated in the text Line 627. The axis label was miswritten, but annual data was correctly displayed (the two datasets are provided in Dätwyler et al., 2018, but we finally selected the annual index to compare with $\delta^{18}O$ and $^{15}N_{excess}$ data which are not season-restricted). We will rectify the axis label.

**"Caption L620 'show'."**

Changed to "Annual resolution is represented by thin lines"

**"Author contribution"**

**"L928. Say something like 'contributed to with comments on the initial manuscript', as otherwise, it sounds like the coauthors were reviewers."**

Changed to "AP, AJ, MC, AM, AL, JMC, ELM, XF, and JC contributed to the redaction with comments, suggestions and corrections on the manuscript."

**Additional corrections**

Line 499: Corrected the citation to Hughes et al., 2021 which was not in the reference list.

Line 801: Added a missing "."

Line 448 cite Kobashi et al., 2015

**references**

Agosta, C., Amory, C., Kittel, C., Orsi, A. J., Favier, V., Gallée, H., van den Broeke, M. R., Lenaerts, J. T. M., van Wessem, J. M., van de Berg, W. J., and Fettweis, X.: Estimation of the Antarctic surface mass balance using the regional climate model MAR (1979–2015) and identification of dominant processes, The Cryosphere, 13, 281–296, https://doi.org/10.5194/tc-13-281-2019, 2019.

Broeke, M. R. V. D. and Lipzig, N. P. M. V.: Response of Wintertime Antarctic Temperatures to the Antarctic Oscillation: Results of a Regional Climate Model, in: Antarctic Peninsula Climate Variability: Historical and Paleoenvironmental Perspectives, American Geophysical Union (AGU), 43–58, https://doi.org/10.1029/AR079p0043, 2003.

Dätwyler, C., Neukom, R., Abram, N. J., Gallant, A. J. E., Grosjean, M., Jacques-Coper, M., Karoly, D. J., and Villalba, R.: Teleconnection stationarity, variability and trends of the Southern Annular Mode (SAM) during the last millennium, Clim Dyn, 51, 2321–2339, https://doi.org/10.1007/s00382-017-4015-0, 2018.

Dee, D. P., Uppala, S. M., Simmons, A. J., Berrisford, P., Poli, P., Kobayashi, S., Andrae, U., Balmaseda, M. A., Balsamo, G., Bauer, P., Bechtold, P., Beljaars, A. C. M., Berg, L. van de, Bidlot, J., Bormann, N., Delsol, C., Dragani, R., Fuentes, M., Geer, A. J., Haimberger, L., Healy, S. B., Hersbach, H., Hólm, E. V., Isaksen, L., Kållberg, P., Köhler, M., Matricardi, M., McNally, A. P., Monge-Sanz, B. M., Morcrette, J.-J., Park, B.-K., Peubey, C., Rosnay, P. de, Tavolato, C., Thépaut, J.-N., and Vitart, F.: The ERA-Interim reanalysis:

configuration and performance of the data assimilation system, Quarterly Journal of the Royal Meteorological Society, 137, 553–597, https://doi.org/10.1002/qj.828, 2011.

Gong, D. and Wang, S.: Definition of Antarctic Oscillation index, Geophysical Research Letters, 26, 459–462, https://doi.org/10.1029/1999GL900003, 1999.

Hughes, A. G., Wahl, S., Jones, T. R., Zuhr, A., Hörhold, M., White, J. W. C., and Steen-Larsen, H. C.: The role of sublimation as a driver of climate signals in the water isotope content of surface snow: laboratory and field experimental results, The Cryosphere, 15, 4949–4974, https://doi.org/10.5194/tc-15-4949-2021, 2021.

Kobashi, T., Ikeda-Fukazawa, T., Suwa, M., Schwander, J., Kameda, T., Lundin, J., Hori, A., Motoyama, H., Döring, M., and Leuenberger, M.: Post-bubble close-off fractionation of gases in polar firn and ice cores: effects of accumulation rate on permeation through overloading pressure, Atmos. Chem. Phys., 15, 13895–13914, https://doi.org/10.5194/acp-15-13895-2015, 2015.

Limpasuvan, V. and Hartmann, D. L.: Eddies and the annular modes of climate variability, Geophysical Research Letters, 26, 3133–3136, https://doi.org/10.1029/1999GL010478, 1999.

Prieto, G. A.: The *Multitaper* Spectrum Analysis Package in Python, Seismological Research Letters, 93, 1922–1929, https://doi.org/10.1785/0220210332, 2022.

Servettaz, A. P. M., Orsi, A. J., Curran, M. A. J., Moy, A. D., Landais, A., Agosta, C., Winton, V. H. L., Touzeau, A., McConnell, J. R., Werner, M., and Baroni, M.: Snowfall and Water Stable Isotope Variability in East Antarctica Controlled by Warm Synoptic Events, J. Geophys. Res. Atmos., 125, https://doi.org/10.1029/2020JD032863, 2020.

---

## Author Comment (AC2)

**Note: Figures in this document are numbered from Fig. R4, following the 3 Figures in the response to Referee #1. Comments from the referee are underlined, and our response follows.**

We thank Referee #2 for their constructive comments. As requested, we will address here the specific comments.

**"Consider separating into paired (or complementary) publications"**

This work is a part of a collaborative research project on the Aurora Basin North ice core. Anais Orsi and I, Aymeric Servettaz, were in charge of the analysis of gases isotopes. However, to interpret the gas isotopes, substantial amount of data is required, such as age models, ice flow, borehole temperature, and comparison with commonly used water isotopes was deemed necessary. While these data could deservedly be published in independent papers, we do not have the priority on the writing of these articles, and try to keep their use to a minimum, although with detailed explanation on innovative methods.

Moreover, following the comments of the two referees, the new manuscript will be simplified and some arguments removed.

For these reasons, we would like to keep all information into one manuscript.

**"Smoothing of gas isotope data"**

In my understanding, trapping heterogeneities result from differential rate of bubble closure and trapping of the gases, which is not mass-dependent but size-dependent (Severinghaus and Battle, 2006), and influences $\delta^{40}Ar$ as argon is more easily expelled due to its shorter radius (Kobashi et al., 2015). For comparison, we represent the $^{15}N_{excess}$ of all samples with and without the 5-m smoothing (Figs. R4 & R5). Without smoothing, the signal is noisy, and we cannot interpret these variations in terms of climate. The uncertainty of each point as well as dispersion of averaged points in the 5-m window is pooled and taken into account for the temperature reconstruction (as described P7 L203). We chose to smooth on a 5-m window to include at least two ice samples from different depths, because a single sample depth is not considered sufficient to represent the average firn gas content due to vertical differences in pore closure rates. Firn densification studies show vertical heterogeneities on scales of a few cm (Hörhold et al., 2011), which could be reflected by close-off heterogeneities of the same scale. We suggest changing the reference to the peer-reviewed article cm (Hörhold et al., 2011) in the sentence: "Due to the high frequency variability of gases, the isotopic composition cannot be related to climate information. The bubble trapping was shown to be heterogeneous at the 10 cm scale, causing variability in the isotopic composition of the gases (Orsi, 2013), probably because of the differential closure rate of bubbles in summer versus winter layers of ice 200 (Severinghaus and Battle, 2006)." replaced by "Vertical heterogeneities in firn density (Hörhold et al., 2011) can lead to differences in bubble closure rate with a size-dependant fractionation (Severinghaus and Battle, 2006), and consequently imprint a high-frequency non-climatic signal in $^{15}N_{excess}$ (Kobashi et al., 2015). To reduce the noise induced by pore closure, we average samples in 5-m windows to include samples from at least two distinct depths."

[Figure]

Figure R4 (added as a subplot of Fig. 2): $^{15}N_{excess}$ in the ABN1314 ice core (defined as $\delta^{15}N - \frac{1}{4}\delta^{40}Ar$), for each depth (light green error-bars show the dispersion for each depth) and with 5-m window averages (dark green line).

[Figure]

Figure R5: $^{15}N_{excess}$ in the ABN1314 ice core, for each depth (light green error-bars show the dispersion for each depth, average of each depth as dark green line).

**"The influence of advection/ice flow on borehole temperature"**

This comment highlighted a mistake on Appendix Fig. A15. We argued in section 4.2 that the "steepening gradient of the temperature above 100 m below surface (Fig. 3)" (Page 26 Line 594) was caused by climatic causes, however the transient-state simulation profile in Appendix Fig. A15 suggested that the borehole temperature profile could be explained by advection only. This was in fact an error where we mistakenly plotted the wrong temperature profile resulting from a previous inversion (i.e., including climate signal to fit to real borehole data), instead of advection only. The Fig. A15 should be replaced by the Fig. R6, where the "steepening of the gradient of temperature" clearly stands out from advection effects of the transient-state simulation. So yes, this can really be seen over the strong advection-based signal.

[Figure]

Figure R6 (replacement for Fig. A15): Temperature profile obtained after the transient state initialization (red dashed line), as compared to the borehole temperature profile (grey points) and the estimated temperature profile after inversion (green), on the upper 300 m (**a**) and the upper 1300 m (**b**) of the ice column.

**"Discussion sections"**

**"gas isotope data are reflecting [...] slope modulated katabatic wind strength and its influence on the strength of the inversion layer and/or convective zone"**

Justification so far was based on timing mismatch of gentle slope and cold periods in the records. We conducted further statistical analysis on the relationship between slope and $^{15}N_{excess}$ (Fig. R7). We will add the following sentences to justify the climatological interpretation of the signal: "Linear regression of reconstructed surface temperature and slope at source ice is 0.24°C (m km$^{-1}$)$^{-1}$ with a squared Pearson correlation **r$^2$** lower than 0.09, which does not support a strong influence of slope on the average surface temperature. At most, the full range of slope variation would explain a difference of 1°C, with low confidence. Therefore, we attribute the changes in $^{15}N_{excess}$ to climate factors rather than advection-related changes of slope."

[Figure]

Figure R7: Scatterplot of $^{15}N_{excess}$ temperature reconstruction against slope at source ice, and linear regression showing a weak negative correlation.

**"SAM specific"**

Discussion sections have been reworked, as described in the response to Referee #1.

**"MINOR COMMENTS"**

**Line 81: Not totally clear what is meant here.**

The sentence "which is why the isotope – temperature slope should be carefully calibrated as close as possible as the expected variability" will be replaced by "which is why the isotope – temperature slope should be carefully calibrated on averaged time-periods as close as possible to the proxy time resolution"

**Line 85-86: … is the main source of (temporal?) variation…**

"temporal" will be added as suggested

**Line 90-92: I'm not clear on the meaning of this sentence.**

"The diffusion of the isotopes of inert gases differs because of their physical properties, with primary control by gravitational settling of heavy gases at the bottom of the diffusive column" replaced by "The

diffusion of inert gases through the firn is accompanied by fractionation of elements and isotopes due to the difference in their physical properties. The primary source of fractionation is the gravitational settling of heavy gases at the bottom of the diffusive column".

**Lines 105-108: This sentence is confusing. Also – is there a reference to cite for the ideal accumulation range for gas isotope-based temperature reconstructions?**

Split and rephrased for clarity. "Accumulation rate controls the closing speed of firn porosity, and thus restricts the locations where this method can be used to infer temperature changes. Low accumulation rates allow time for the firn ice matrix to equilibrate its temperature with the surface before the porosity is closed, minimizing the firn temperature gradient that can be captured in the gases isotopes. High accumulation rates do not allow time for gases to diffuse through the firn and equilibrate with the temperature gradient, so the gases isotopes do not record the full extent of temperature changes. Therefore, this method has been applied for sites with accumulation rates between 74 kg m$^{-2}$ yr$^{-1}$ (South Pole, Morgan et al., 2022) and 220 kg m$^{-2}$ yr$^{-1}$ (GISP2, Kobashi et al., 2015)"

**Line 122: Can you include a site mean annual temperature here?**

We will add "and the annual mean temperature is estimated at -42.0°C (Automatic Weather Station, 2015 to 2021 average)"

**Line 153: Approximately how much ice was shaved off?**

We will precise "The samples' outer 5-mm layer of ice was shaved off to prevent contamination…"
**Line 164: … laboratory standard of (combined?) N2, Ar, (and) Kr.**

We will precise "laboratory standard gas mixture of N2, Ar, Kr"

**Line 172: Which elemental ratios were measured? If none are shown, maybe not a necesary detail to include.**

In practice we measured Ar/N2 and Kr/Ar ratios, but for this study we only use Ar/N2 to confirm the quality of ice (no $\delta^{40}$Ar $- \delta$Ar/N2 correlation). We will precise the following sentences:

Line 172 "Additionally, elemental ratios of Ar/N2 were measured following the peak-jumping method (Bereiter et al., 2018)."

Line 183 "the excellent quality of ice from a recently drilled ice core, and the precautions taken during the preparation prevented any notable effect of argon loss during storage on the δ40Ar measured in our samples, attested by the absence of correlation between $\delta^{40}$Ar and $\delta$Ar/N2"

**Line 180: Please include the original and improved pooled standard deviations.**

We precise "This drift correction reduced the pooled standard deviation of δ40Ar in the ice duplicates from 0.028 ‰ to 0.013 ‰."

**Line 195: … on thinly closed (pores?) may have…**

Changed "porosity" to "pores".

**Line 201-202: This is a bit strongly worded here - I would suggest adding some caveats to this statement.**

This sentence will be removed as the paragraph is rephrased as indicated in the response to major comments.

**Line 211: How long was the probe equilibrated for?**

We will precise: "The probe was left to equilibrate at each depth interval such that the read-out was verified as unchanging. This was achieved within a few minutes and then left to equilibrate an additional 3 to 5 minutes to ensure a stable value."

**Lines 213-215: Unclear how this would work.**

There was a confusion between the start of wet drilling (132 m, depth at which ice drilling started with fluid) and the fluid level (up to approximately 100 m after ending the drilling). We will correct the text as follows: replace "Wet drilling (Estisol) commenced from 132 m, and it is very likely that the open markers in Fig. 3 are outliers due to disturbance of air in the drill hole with warm fluid stored at the surface. Below 132 m, the small difference between upward and downward measurements is likely due to improved 215 equilibrium in the drilling fluid." by "The temperature disturbance at ~100 m depth is attributable to the addition of drilling fluid (Estisol) stored at the surface into the drill hole, with the last addition just a few days before temperature profiling. Open markers in Fig. 3 will be considered as outliers for this reason."

**Line 260: How many tiepoints?**

We will add Line 264: "Fourteen tie points were identified where there is clear, quick transitions or extrema on methane records (Fig. A2)." And "For the most recent part […] the ABN methane record was tied to the revised Law Dome record (Rubino et al., 2019) with 4 additional tie points".

**Figure 5: Could the last part of figure A3 also be shown here? This is a neat (but somewhat complicated) way to reconstruct ice flow, so it may make sense to either add some of the details from FigA3 here or move everything to the appendix.**

The ice-flow correction requires long justification by first retrieving flowrate from accumulation, but also the elevation at source ice as well as a calibration of spatial $\delta^{18}$O slope for this region which we found too long to detail in the main text. Most justifications are in the appendix, but we thought that showing the $\delta^{18}$O values was important here, because in the remainder of the article we discuss the temperature reconstructed from $\delta^{18}$O, and not the $\delta^{18}$O directly. We think that showing the $\delta^{18}$O at least once in the main body is better. We thus suggest keeping the current structure, but remain open to change if it is deemed necessary.

**Line 394: It might make it more clear if you use 'pooled standard deviation' rather than 'difference' here. Also – it would be informative to see d15Nexcess plotted in figure 2 (as a subplot) with both the individual values and moving means.**

We will use "pooled standard deviation" rather than "difference" as suggested. Fig. R4 will be added in Figure 2 as a subplot.

**Figure 7: There is no blue shading in this figure (only red). Also – the authors may have flipped the y-axis firn column height to show similarities with the temperature profile but this isn't clear in the caption and isn't how this information is normally presented. One last suggestion – given that the authors suggest that the lock in depth is a function of ice accumulation (line 423), it might make sense to show reconstructed accumulation (as shown in A3) in this figure.**

There was a compilation bug where the blue shading was hidden during the creation of PDF for submission. Sorry for not picking up this mistake, the actual figure will be Fig. R8. We will further describe: "Y-axis for the firn column depth was flipped so that deeper lock-in depths are represented as lower points." I think it is common for coring studies or oceanographic studies to represent deeper points below, with the top of axis being the minimum depth. For ease of reading, we can add arrows pointing shallower and deeper firn.

The representation of Accumulation in this figure is possible (Fig. R9) but we think it is better not to show here to avoid overcomplexity of the figure which could drive the attention away from its original intent. Instead, we will add reference to studies showing this well-known mechanism "(Sowers et al., 1992; Goujon et al., 2003)".

[Figure]

Figure R8 (to be included as a replacement of Fig. 7): (a) Series of ΔT computed from 15Nexcess. Orange shadings indicate a warming (ΔT>0) and blue shadings a cooling (ΔT<0). (b) Past lock-in depth (thick grey line) estimated from diffusive column height of gases isotopes (black line with error bars) and gas-410 ice depth difference (blue dashed line). Yellow shadings highlight the potential presence of a convection zone that would be located in the uppermost layer of the firn (0~5 m depth), when the lock-in depth

appears to be deeper than the diffusive column height. For clarity, uncertainties on the lock-in depth are only shown at both ends of the record. White dots on the lock-in depth indicate the ages where the gas age model was tied to WD2014, indicating the constraints on the Δdepth.

[Figure]

Figure R9: same as Fig. R8, with addition of ABN1314 accumulation derived from the annual layer counting and density measurements. We argue that this link was shown in previous studies as is not necessary to show in the article, to keep the focus on other points discussed.

**Lines 470-472: For those unfamiliar with inversions, it's unclear what this means or why it's being stated here.**

The formulation "We set the inversion to use an exponentially decreasing covariance in the linear combination, which reaches 0.5 for a time difference of 70 year, roughly twice the time resolution of the gas constraints on ΔT." was extremely unclear. We will rewrite:

"During the inversion, we use a smoothing parameter to avoid noisy reconstruction with sharp, unrealistic transitions. Temperature points in the inversion are forced into a limited range, determined as an exponentially decreasing tie to neighbor points so that two points at a time difference of 70 years have a covariance of 0.5. This window ensures that each point of the inversion is influenced by gas constraints on ΔT, which have an average time resolution of 45 years."

**Figure 9 (and others): I might have missed it, but I'm not sure the 'way out' and 'way back' on this figure was ever explained.**

I omitted to describe upstream GPS data. Captions of Figs. 9 and A3 will be completed by "Elevation was determined with truck GPS position during upstream radar profiling. The profile was taken twice: moving away from coring site (way out) and going back to coring site (way back). Original GPS coordinates were not taken in optimal conditions (moving truck), hence the uncertainty."

**Line 595-597: The slope still appears relatively steep during this interval.**

This describes the period after 1900 CE, when the slope is changing from -2 m km$^{-1}$ to -1 m km$^{-1}$. Although it is still not flat, the slope's absolute value is decreasing while the temperature is increasing. Therefore, we interpret the temperature change as a climatic warming, because this flattening "would on the contrary favour the slowing of katabatic winds and surface cooling by strengthening of the near-surface temperature inversion".

**Figure A7: What are the units on the x-axis here? Millivolts?**

Yes, the beam intensity in faraday cups is converted to mV tension. We will correct the axis labels accordingly.

**Figure A1: Looks like the legend has a typo – offsets are larger than what was actually applied.**

The caption was missing a plus/minus sign We will correct to "±5-year uncertainty", as correctly stated in text line 243.

**References**

Goujon, C., Barnola, J.-M., and Ritz, C.: Modeling the densification of polar firn including heat diffusion: Application to close-off characteristics and gas isotopic fractionation for Antarctica and Greenland sites: A NEW FIRN DENSIFICATION MODEL, J. Geophys. Res., 108, n/a-n/a, https://doi.org/10.1029/2002JD003319, 2003.

Hörhold, M. W., Kipfstuhl, S., Wilhelms, F., Freitag, J., and Frenzel, A.: The densification of layered polar firn, Journal of Geophysical Research: Earth Surface, 116, https://doi.org/10.1029/2009JF001630, 2011.

Kobashi, T., Ikeda-Fukazawa, T., Suwa, M., Schwander, J., Kameda, T., Lundin, J., Hori, A., Motoyama, H., Döring, M., and Leuenberger, M.: Post-bubble close-off fractionation of gases in polar firn and ice cores:

effects of accumulation rate on permeation through overloading pressure, Atmos. Chem. Phys., 15, 13895–13914, https://doi.org/10.5194/acp-15-13895-2015, 2015.

Morgan, J. D., Buizert, C., Fudge, T. J., Kawamura, K., Severinghaus, J. P., and Trudinger, C. M.: Gas isotope thermometry in the South Pole and Dome Fuji ice cores provides evidence for seasonal rectification of ice core gas records, The Cryosphere, 16, 2947–2966, https://doi.org/10.5194/tc-16-2947-2022, 2022.

Severinghaus, J. and Battle, M.: Fractionation of gases in polar ice during bubble close-off: New constraints from firn air Ne, Kr and Xe observations, Earth and Planetary Science Letters, 244, 474–500, https://doi.org/10.1016/j.epsl.2006.01.032, 2006.

Sowers, T., Bender, M., Raynaud, D., and Korotkevich, Y. S.: δ15N of N2 in air trapped in polar ice: A tracer of gas transport in the firn and a possible constraint on ice age-gas age differences, Journal of Geophysical Research: Atmospheres, 97, 15683–15697, https://doi.org/10.1029/92JD01297, 1992.

---

## Author Response (AR1)

Author Response, File upload (CP) Iteration: Correction

Dear Liz Thomas,

I applied the changes discussed in the responses to Referees, and uploaded the corrected files. I carefully checked each figure for any compilation issue. Thank you for your work dedicated to this article.

Note for the person in charge of the layout: Following the recommendations for text files, I moved all figure captions at the end of the file. Some Appendix sections do not include text so it would be easier to read if figures were anchored in the corresponding Appendix section, as in the manuscript pdf.

Sincerely yours,

Aymeric Servettaz